



# The Arctic Weather Satellite radiometer

Patrick Eriksson[6], Anders Emrich[1], Kalle Kempe[1], Johan Riesbeck[1], Alhassan Aljarosha[1], Olivier Auriacombe[1], Joakim Kugelberg[2], Enne Hekma[2], Roland Albers[3], Axel Murk[3], Søren Møller Pedersen[4], Laurenz John[5], Jan Stake[6], Peter McEvoy[6], Bengt Rydberg[7], Adam Dybbroe[7], Anke Thoss[7], Alessio Canestri[8], Christophe Accadia[8], Paolo Colucci[8], Daniele Gherardi[9], and Ville Kangas[9]

[1]Omnisys Instruments AB, Västra Frölunda, Sweden
[2]OHB Sweden AB, Kista, Sweden
[3]Institute of Applied Physics, Bern University, Bern, Switzerland
[4]Technical University of Denmark, Kongens Lyngby, Denmark
[5]Fraunhofer Institute for Applied Solid State Physics, Freiburg, Germany
[6]Chalmers University of Technology, Gothenburg, Sweden
[7]Swedish Meteorological and Hydrological Institute (SMHI), Norrköping, Sweden
[8]EUMETSAT, Darmstadt, Germany
[9]European Space Agency/ESTEC, Noordwijk, the Netherlands

**Correspondence:** Patrick Eriksson (patrick.eriksson@chalmers)

**Abstract.** The Arctic Weather Satellite (AWS) is a project led by the European Space Agency (ESA) that has several novel aspects. From a technical perspective, it serves as a demonstrator of how to expand the network of operational satellite-based microwave sensors cost-effectively and acts as the proto-flight model for a suggested constellation of satellites, denoted as EUMETSAT Polar System (EPS) Sterna. The design philosophy has been to reduce complexity and instead focus the efforts on

critical parts and characterise the instrument well before the launch. The single instrument onboard is a 19-channel microwave cross-track radiometer. There are 15 channels covering ranges around 54, 89 and 174 GHz. These are channels similar to ones found on existing sensors, however, thanks to the short development process, allowing use of more modern and recent technology, the performance and resolution of these channels on AWS exceed or match similar sensors, despite being a small satellite. Additionally, four channels around 325.15 GHz form a completely new frequency band for observations from space.

The addition of these new channels aims to improve sensitivity to ice hydrometeors.

   In this article, we outline the mission and describe the instrument to support the usage of radiances measured by AWS. The satellite was launched in August 2024, and the status towards the end of the commissioning phase is reflected here. For example, a characterisation of the noise performance is provided, showing that the target specifications have been met, for most channels with a margin. This is except for two channels identified to have technical issues already before the launch. If

EPS-Sterna is selected by EUMETSAT, these and other identified problems will be corrected, but otherwise the constellation is expected to consist of recurrent models of AWS with minor modifications.





## 1 Introduction

Observations from space are essential for both global and regional weather forecasting. Satellites used for these purposes fly in geostationary or polar orbits, each with complementary characteristics. A geostationary platform provides continuous data, but only for the area facing the satellite, leaving the polar regions out of reach. The sensors in geostationary orbit are limited to passive optical and infrared wavelength regimes; microwave sensors are still lacking due to the large antenna size required, considering the high orbit altitude (about 36,000 km). A more diverse set of sensors is found on polar-orbiting satellites. Most of these instruments provide close to global coverage daily, but most locations are only observed at two fixed local solar times as sun-synchronous orbits are preferred.

Improvements in weather forecasts can be achieved in several ways, such as increased computing power, better assimilation procedures, and more or new observations (Bauer et al., 2015). The developments in these areas are not independent. For instance, introducing all-sky assimilation (not rejecting cloud-affected radiances) significantly enhanced the relative importance of microwave channels sensitive to humidity in forecasts (Geer et al., 2017). Besides having a high general impact on forecasts, microwave observations are particularly important for mapping severe weather (Boukabara et al., 2007). Such weather systems are associated with deep, thick cloud decks, and only sensors operating at long wavelengths offer sensitivity to air volumes below the cloud top layer. However, in such cases and others, the space system must provide microwave observations with considerably higher temporal sampling to fully utilise what this wavelength region can offer.

The Metop (operated by EUMETSAT, www.eumetsat.int/metop) and JPSS (NOAA, www.noaasis.noaa.gov/POLAR/JPSS/jpss.html) programs are leading examples of polar orbiting weather satellites. They exemplify the current approach of creating series of large platforms carrying multiple sensors. For instance, the Metop satellites are equipped with up to eight weather instruments (AMSU-A, ASCAT, AVHRR, GOME, GRAS, HIRS, IASI, and MHS), among other systems, on the same platform. Consequently, these platforms offer co-located observations using various techniques and different parts of the electromagnetic spectrum. While providing joint observations is not a requirement for current assimilation systems, it can still be beneficial for research purposes. However, this relatively marginal advantage comes at a high cost, as multiple-sensor platforms are complex to develop and require long implementation phases.

The Arctic Weather Satellite (AWS) is a mission based on a different design philosophy. It is constructed around a moderately sized platform and carries a single instrument, a cross-track microwave radiometer. The ambition has been to develop and launch an instrument of operational standard, but significantly faster and at a lower cost than traditionally achieved. The satellite was launched in August 2024. No extension of either the financial budget or time plan was needed. As a consequence of rapid development, the technology in AWS still represented the state-of-the-art at the time of launch.

The AWS radiometer (Fig. 1) has 19 channels. Most of these channels correspond to those found on currently flown microwave sounders and imagers between 50 and 190 GHz, providing information on atmospheric temperature and humidity profiles. To maintain a small instrument volume, channels below 50 GHz were not included. Additionally, AWS has four channels around 325.15 GHz, making it the first operational sensor to measure at sub-millimetre wavelengths (i.e. above 300 GHz). These channels are suitable for measuring properties of ice hydrometeors (Evans et al., 2005).




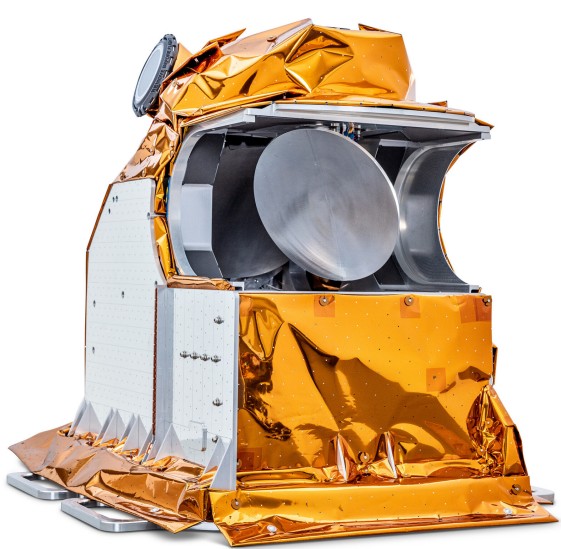

**Figure 1.** Photo of the Arctic Weather Satellite payload, a 19-channel microwave cross-track sounder. The metallic plates on the left side are thermal radiators. Above these, one of the two star trackers can be discerned. The scanning is achieved by the elliptical mirror, in the photo seen pointing to the left. The mirror is attached to an axis above it and directing radiation to the radiometer package below it. The front side will in space be facing the nadir direction.

AWS will be used as an operational mission, but also acts as a demonstrator for achieving a constellation of satellites, the EUMETSAT Polar System Sterna (EPS-Sterna) aiming at providing microwave radiances with unprecedented coverage and re-visit time. If funded (to be decided, by EUMETSAT), the planned nominal constellation will consist of six satellites distributed over three orbital planes complementing the existing IJPS (International Joint Polar Systems) satellites. The combination of
EPS-Sterna and IJPS will provide a temporal sampling of Europe that, in general, is better than half an hour (Varley, 2023; Rivoire et al., 2024). During the full extent of the EPS-Sterna program, the launch of 18 satellites is foreseen, with a planned start 2029. The EPS-Sterna constellation satellites will be recurrent models, with minor modifications, of the AWS, denoted by the European Space Agency (ESA) as the proto-flight model (PFM).

The following sections give an introduction to the AWS PFM, including the information needed to make proper use of
measured radiances for data assimilation and other forms of retrievals. Sec. 2 introduces the radiometer onboard AWS, followed by a brief overview of the technical aspects of the overall mission in Sec. 3. The on-ground characterisation of the AWS instrument is summarised in Sec. 4, and some aspects of the AWS data are illustrated in Sec. 5 based on simulations. Sec. 5 also contains examples of actual AWS measurements and an initial assessment of the instrument's noise characteristics.



## 2 The radiometer

### 2.1 Background


The AWS mission emerged from the recognised need to extend, in a cost effective manner, the set of space-based microwave radiometers used for weather forecasting. The starting point was a concept for a small microwave sounder developed between 2016 and 2019 at Omnsisys Instruments AB (AAC Omnsys), the instrument developer and manufacturer based in Gothenburg, Sweden. Based on input from individuals at various weather agencies (SMHI, UK Met Office, EUMETSAT and ECMWF), the

concept addressed, in particular, the need for a better temporal coverage of measurements around 183 GHz. A constellation of sounders was envisaged from the start. A secondary aim was to offer 10 km nadir resolution, compared to about 15 km as the best resolution found among other microwave sounders (e.g. ATMS, Kim et al. (2014)). This higher resolution would better align with regional and future global atmospheric models, which will have a spatial resolution of just a few kilometres.

The concept was extended to a complete satellite constellation together with OHB Sweden, based on their Innosat platform

(Lagaune et al., 2021). This work was part of an ESA study exploring the needs of new satellite systems to better support services in the Arctic and surrounding regions (OHB Sweden and Thales Alenia Space, 2019). Based on the mission plan developed, the Swedish National Space Agency proposed the Arctic Weather Satellite to the ESA council meeting in November 2019, and the project received broad support.

ESA took on the project with a non-standard approach, and as a result, AWS is ESA's first satellite developed using a "new

space" philosophy. The usual standards were not applied. Instead ESA identified a subset of critical product assurance and performance requirements, that were applied by the industry team. The project management largely followed Agile principles, with a focus on rapid development by frequent testing in order to quickly identify critical problems. The contract between ESA and the industrial consortium was signed in March 2021; the instrument was provided for integration with the platform 31 months later, with the entire satellite ready and tested after five additional months.


In parallel, EUMETSAT defined the user requirements for EPS-Sterna, conducted scientific impact studies (Guedj et al., 2023; Perrels and Juhanko, 2023; Rivoire et al., 2024; Lean et al., 2025; Rydberg et al., 2024) and a socio-economic benefit assessment for the EPS-Sterna constellation (Varley, 2023), to prepare the programme approval, pending on the in-flight performance of AWS-PFM.

### 2.2 Basic design

A first choice was to use a cross-track sounder, as this instrument type allows for a more compact design compared to a conically scanning instrument. To achieve 10 km resolution at 183 GHz, a main reflector having an effective diameter of about 0.16 m is needed, becoming the main design driver. An altitude of 600 km has been targeted as a good compromise between antenna size requirement, ground coverage, launch cost and end-of-life deorbiting considerations.

To complement the sounding channels around 183 GHz, the standard window channels at 166 and 89 GHz were included in

the design. The 183 and 166 GHz channels were combined in a novel manner into a single receiver chain (below denoted as the 174 GHz receiver), while the 89 GHz channel still needed to be implemented separately to meet the end user requirements.





To minimise the instrument volume and simplify the overall design, spatial separation was chosen for the accommodation of the receivers. Due to this separation, a notable property of the AWS radiometer is that the footprints of the channel groups are not fully collocated, as the feed-horns look into the main reflector from slightly different directions. However, with this decision, it was concluded that accommodating four receiver chains was possible with relative ease if they were made compact. Based on the study by Lean et al. (2022), the 54 GHz range for temperature sounding was added to the concept. Up to eight channels were deemed appropriate to be handled by this third receiver chain, based on the initial pre-development work at AAC Omnisys.

The MicroWave Sounder (MWS, Kangas et al. (2012)), to be part of the three satellites in the Metop second generation (Metop-SG) series, was used as the starting point for the detailed channel definitions, with some exceptions. While MWS will have channels around 23.8 and 31.4 GHz, these frequencies were excluded as they need a larger reflector size. Compared to MWS, AWS does not include the most high-altitude temperature channels. Both the 54 and 174 GHz AWS receiver chains operate fully in a single-sideband fashion, while MWS has e.g. side-bands also on the high-frequency side of the water vapour 183.31 GHz transition.

Further, present operational microwave sensors have only channels below 190 GHz, but MWS will take the step to 229 GHz to increase the sensitivity to ice hydrometeors. This channel was considered for AWS, but after a study financed by EUMET-SAT, it was decided to instead include four channels around 325.15 GHz as this set of channels offer a better basis for both cloud filtering and retrievals (Kaur et al., 2021). The 325.15 GHz band will also be used by another upcoming Metop-SG sensor, the Ice Cloud Imager (ICI, Eriksson et al. (2020)).

The AWS radiometer is designed to measure Earth antenna temperatures between 80 and 315 K. The overall instrument design allows for configuration of any new deployments instrument design to comply with almost any sun-synchronous orbit. All sensitive electronic parts are placed centrally, and radiators can be moved from one side of the instrument to the other with minimal impact on the thermal range and stability of critical parts. Control and power sections are fully redundant.

The target values for the implementation of the 19 AWS channels are found in Table 1, and Fig. 2 provides an overview of the components of the actual radiometer.

## 2.3 Receiver chains

As outlined above, there are four receiver chains, covering the channels around 54, 89, 174 and 325 GHz, respectively. Their feed-horns are all smooth-walled spline horns (Hammar et al., 2016), arranged in a split-block. The polarisation response of the four receivers is aligned, see further Sec. 2.6.

Most of the control, power distribution and bias components have been standardised between the chains, to reduce development time and cost. For the same reason, synthesised local oscillators based on heterodyne receivers were selected (in favour over dielectric resonator oscillators). This choice is also advantageous with respect to sensitivity to temperature changes and ageing. To minimise losses affecting NE$\Delta$T (noise equivalent differential temperature), there are no optical components between the receiver horns and the main reflector.





| Channel | Frequency [GHz] | Bandwidth [GHz] | Footprint FWHM [km] | NEΔT [K] | Integration time [ms] | Polari- sation |
|---|---|---|---|---|---|---|
| AWS11 | 50.300 | 0.18 | < 40 | < 0.6 | 10.0 | QV |
| AWS12 | 52.800 | 0.40 | < 40 | < 0.4 | 10.0 | QV |
| AWS13 | 53.246 | 0.30 | < 40 | < 0.4 | 10.0 | QV |
| AWS14 | 53.596 | 0.37 | < 40 | < 0.4 | 10.0 | QV |
| AWS15 | 54.400 | 0.40 | < 40 | < 0.4 | 10.0 | QV |
| AWS16 | 54.940 | 0.40 | < 40 | < 0.4 | 10.0 | QV |
| AWS17 | 55.500 | 0.33 | < 40 | < 0.5 | 10.0 | QV |
| AWS18 | 57.290 | 0.33 | < 40 | < 0.6 | 10.0 | QV |
| AWS21 | 89.000 | 4.00 | < 20 | < 0.3 | 5 | QV |
| AWS31 | 165.500 | 2.80 | 10 | < 0.6 | 2.5 | QV |
| AWS32 | 176.311 | 2.00 | 10 | < 0.7 | 2.5 | QV |
| AWS33 | 178.811 | 2.00 | 10 | < 0.7 | 2.5 | QV |
| AWS34 | 180.311 | 1.00 | 10 | < 1.0 | 2.5 | QV |
| AWS35 | 181.511 | 1.00 | 10 | < 1.0 | 2.5 | QV |
| AWS36 | 182.311 | 0.50 | 10 | < 1.3 | 2.5 | QV |
| AWS41 | $325.150 \pm 1.2$ | $2 \times 0.80$ | 10 | < 1.7 | 2.5 | QV |
| AWS42 | $325.150 \pm 2.4$ | $2 \times 1.20$ | 10 | < 1.4 | 2.5 | QV |
| AWS43 | $325.150 \pm 4.1$ | $2 \times 1.80$ | 10 | < 1.2 | 2.5 | QV |
| AWS44 | $325.150 \pm 6.6$ | $2 \times 2.80$ | 10 | < 1.0 | 2.5 | QV |

**Table 1.** Target characteristics for the Arctic Weather Satellite's radiometer channels. The channels of groups 1-3 (AWS11 to AWS36) are single sideband, while the group 4 channels (AWS41 to AWS44) are double side-band. The target NEΔTs are defined for the integration times listed, and were expected to be met over the full range of antenna temperatures. The level 1b data delivered to end-users will throughout be for the basic integration time of 2.5 ms. The concept of QV (quasi-vertical) polarisation is described in Sec. 2.6.





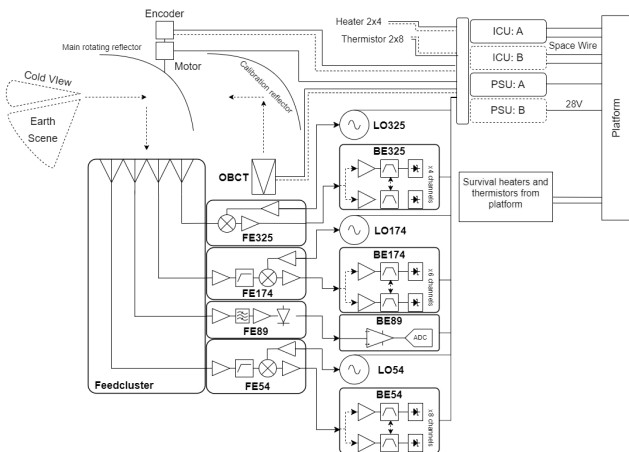

**Figure 2.** Block diagram of the AWS radiometer (FE: front-end, LO: local oscillator, BE: back-end, OBCT: onboard calibration target, ICU: integrated control unit, and PSU: power supply unit).

The 54 GHz receiver chain consists of a single sideband heterodyne front-end (AAC Omnisys), a local synthesised oscillator unit (DA Design Oy) and a filter bank back-end (DA Design Oy). The front-end incorporates a RF low noise amplifier (LNA, Fraunhofer IAF), waveguide filter, a mixer circuit based on Schottky barrier diodes (Chalmers University of Technology (CUT), Drakinskiy et al. (2013)) and amplifier/multiplier components. The LO reference frequency is 12 GHz. The back-end incorporates eight filter channels between intermediate frequencies (IFs) 2.210 and 9.455 GHz.

The 89 GHz receiver (AAC Omnisys and ACST GmbH) is a direct detection single channel implementation, incorporating two LNAs (John et al., 2023), a bandpass waveguide filter and a diode detector supporting video amplification.

The 174 GHz receiver chain consists of a single sideband heterodyne front-end (AAC Omnisys) and a filter bank back-end (AAC Omnisys). The front-end incorporates a RF LNA (Fraunhofer IAF), waveguide filter a mixer circuit based on Schottky diodes (CUT, Anderberg et al. (2019)), amplifiers and multiplier components. The overall IF bandwidth is 20 GHz with a wide low IF (1.525 to 4.325 GHz, 95% fractional bandwidth) and a narrow high IF filter (19.486 to 19.986 GHz, 2.5% fractional bandwidth).

The 325 GHz receiver is based on a traditional Schottky dual sideband mixer topology (Radiometer Physics GmbH) and a filter-bank back-end of four channels, covering 0.8-8.0 GHz (DA Design Oy). The front-end incorporated the Schottky mixer, multipliers and IF amplification stage. The 174 GHz and 325 GHz receivers make use of the same LO source at 13.54 GHz, but in two distinct units (DA Design Oy) .

## 2.4 Onboard calibration target

The AWS onboard calibration target (OBCT) was specified by AAC Omnisys. In contrast to the common design of periodic pyramidal arrays, the AWS OBCT consists of a single wedge shaped cavity. Wedge cavities have a preferred linear polarisation (perpendicular to the wedge apex) for optimal performance, but since the AWS channels all share the same polarisation this




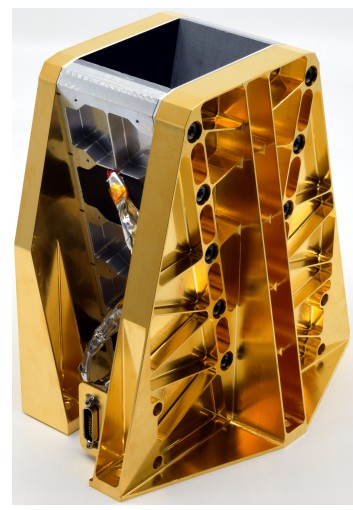

**Figure 3.** The Arctic Weather Satellite onboard calibration target.

has no averse effects. The detailed design and construction was completed by the Institute of Applied Physics (IAP), Bern University. The aperture of the OBCT is rectangular and allow for a 10° scan arc for each band.

Figure 3 shows a picture of the AWS OBCT. The absorber is an epoxy based mixture developed by IAP, which has improved thermal conductivity and is easier to work with than previous mixtures. It is cast on thin aluminium backing plates which form the wedge. The structural shape of the OBCT is driven by the available footprint ($170 \times 110\,\text{mm}^2$) and weight-saving pockets have been used to keep the mass below $1\,\text{kg}$.

A total of eight resistance temperature detectors (RTDs) are potted in the pockets of the aluminium backing plates at two different heights. The RTDs are split into two redundant sets of four, with both sets being spread across both wedge sides to be able to detect both vertical and horizontal temperature gradients. For more details regarding the OBCT design, see Albers et al. (2024b).

## 2.5 Quasi-optical system

There are no mirrors between the feed-cluster and the main reflector (Fig. 4). The latter is an off-axis parabolic mirror with a focal point of $161\,\text{mm}$. The antenna beam towards the Earth is to a first order circular. The mirror is rotating with a (tunable) constant angular speed and is directly attached to the main axle of the motor, saving space and cost as well as potentially improving reliability. The encoders on the antenna axis, used for rotation speed and position control, are redundant. The motor is the only potential single point failure in the instrument, but the windings are oversized to reduce risk.

The scanning mirror provides views towards Earth, cold sky and in the zenith direction the beams are directed into the OBCT. The coupling with the internal calibration target is achieved by an off-axis parabolic mirror with a super-elliptical rim shape projected to maximize the beam efficiency towards the OBCT. A circular cap of the structure is included to redirect the primary spillover lobe of the scanning mirror. The cap is angled such that it reflects the incident spillover past the satellite





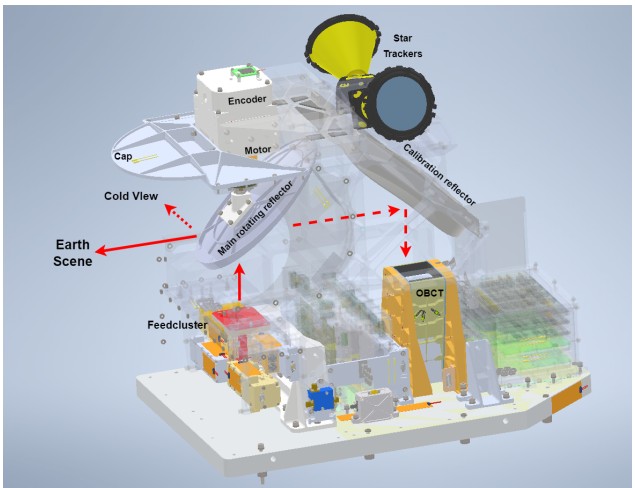

**Figure 4.** Cutaway drawing of the AWS radiometer, focusing on the quasi-optical system and the mounting of star trackers onto the instrument's structure.

into cold space, avoiding Earth incidence. Results of simulations and optimisation of the optical elements are found in Albers et al. (2023). Despite optimisation efforts to minimise spillover, there are still small variations which need to be considered for accurate calibration (Albers et al., 2024a).

## 2.6 Scan sequence

The rotational frequency of the main reflector is 0.84 Hz (50.4 rpm). The scan rate is constant, giving an along-track distance between footprints of about 9.0 km. Inside a rotation, 145 samples towards the Earth, 15 against the OBCT and 25 against the cold sky are recorded. The Earth samples are distributed roughly between scan angles of $\pm 55°$, resulting in a swath width exceeding 2000 km.

A sampling time of 2.5 ms is used throughout, and data are kept at this resolution in level 1b (Sec. 3.4). As the instrument design allows for comparably long observations against cold sky and OBCT, the calibration data will have comparably low noise. For calibrating a swath, the aim is to only include cold sky and OBCT data recorded just before and after the Earth-view (Sec. 3.4). As a consequence, the impact of gain variations at time scales above 2 s is effectively removed.

As mentioned, the beams of the receiver chains are not co-aligned, a design choice to allow for a compact instrument. This results in the beams diverging from the scanning mirror's normal and thus rotate around it as a function of scan angle. Fig. 5 illustrates this behaviour. Although this complicates geolocation, the behaviour is well understood (Albers et al., 2024a). The azimuth and elevation angles of the boresight of the channel groups are given by Eq. 2 and 3 in the aforementioned paper, with the elevation angles shown in Fig. 6 as an example.





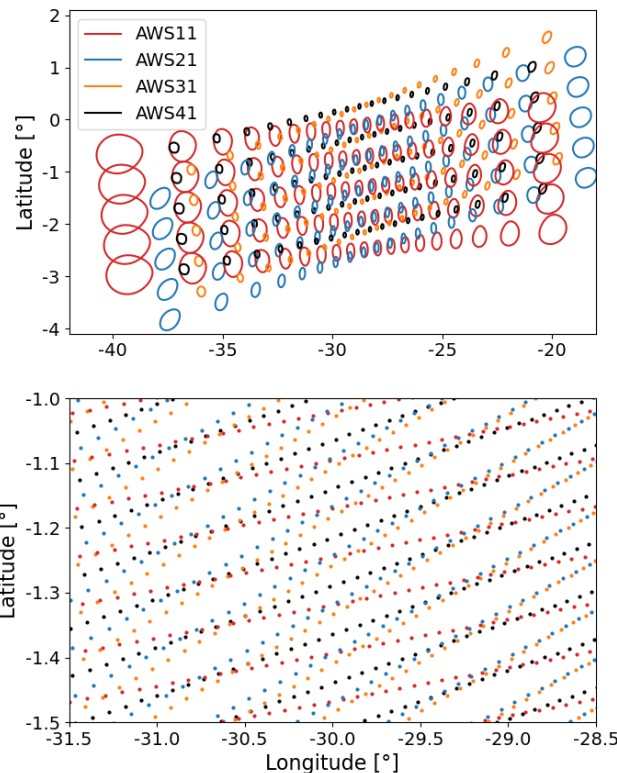

**Figure 5.** AWS' scan pattern. The top panel exemplifies the measurements obtained in about 38 s, but only showing every 8th swath and across-track position to not clutter the figure. Each footprint is displayed as the -3 dB contour of the antenna responses reported in Sec. 4.3. The lower panel shows all boresight positions for a central part of the orbit section.

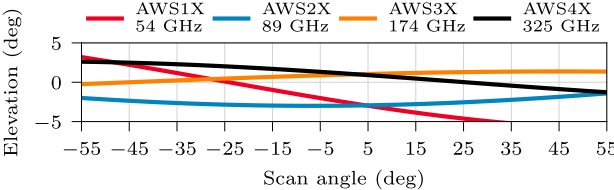

**Figure 6.** The elevation angle (angle from scan plane) of the boresight of the four channel groups, according to Eq. 3 of Albers et al. (2024a).

Alternatively, the unit vector describing the projection of the feed-horns boresight direction, in a sensor head (SH) coordinate system, is (Rydberg, 2024)

$$\boldsymbol{r}_{\mathrm{SH}}(t) = \mathbf{R}(-\theta(t))\boldsymbol{r}_{\mathrm{B}}, \tag{1}$$





| Group | $\theta_i$ | $\phi_i^0$ |
|-------|-----------|-----------|
| AWS1X | 6.12° | 244.04° |
| AWS2X | 3.04° | 174.23° |
| AWS3X | 1.35° | 45.58° |
| AWS4X | 2.63° | 295.96° |

**Table 2.** Reference angles of the four channel groups. From Table 58 in Kempe (2025).

where $t$ is time, $\mathbf{R}$ describes a rotation around the x-axis,

$$\mathbf{R}(\beta) = \begin{bmatrix} 1 & 0 & 0 \\ 0 & \cos(\beta) & \sin(\beta) \\ 0 & -\sin(\beta) & \cos(\beta) \end{bmatrix}, \tag{2}$$

$\theta$ is the mirror angle (pointing towards nadir at time $t = 0$),

$$\theta(t) = 2\pi t/\Delta t, \tag{3}$$

with $\Delta t$ as the time for one full rotation (about $1.191\,\mathrm{s}$), and the vector $\boldsymbol{r}_{\mathrm{B}}$ is defined as:

$$\boldsymbol{r}_{\mathrm{B}} = [\sin(\theta_i)\cos(\phi_i(t)),\ \sin(\theta_i)\sin(\phi_i(t)),\ \cos(\theta_i)]^T, \tag{4}$$

where

$$\phi_i(t) = \phi_i^0 - 2\pi t/\Delta t. \tag{5}$$

The reference angles $\theta_i$ and $\phi_i^0$ are specific for each channel group and are listed in Table 2. In this (right-handed) SH coordinate system, the x-axis is along the local flight direction, the z-axis is the local zenith direction and the y-axis is on the left side when facing the flight direction.

At nadir, the full width at half maximum (FWHM) of the footprints is around 32, 18, 10 and $11\,\mathrm{km}$ for the four receiver chains. More detailed values are found in Sec. 4.3. Just beside nadir, all channels have a vertical polarisation response, a configuration often denoted as "quasi-vertical" (QV). The polarisation response, in the Earth system, changes linearly with mirror angle and, e.g. the mean between vertical and horizontal polarisation is measured at scan angles of 45° away from nadir.

## 3 The mission, in brief

This section covers other technical information of the mission, with focus on data dissemination and formats.

### 3.1 The platform

The AWS satellite is based on the Innosat platform developed by OHB Sweden. This is a three-axis stabilised spacecraft, offered in various configurations. In the case of AWS, including the radiometer, the volume is about $0.6\,\mathrm{m}^3$ (excluding solar





panels), the mass is about 125 kg and the total power consumption is around 120 W. The power is generated by deployable, fixed-angle solar arrays. An L-band (1.707 GHz) radio transmitter provides both a direct broadcast and a stored data downlink capability (Sec. 3.3).

Electric propulsion was added to the platform, to adjust and maintain its orbit and for any collision avoidance manoeuvres that may be needed. The satellite's attitude is determined by two star trackers and controlled by reaction wheels. The star

trackers are placed on the payload structure (Fig. 4) to minimise pointing errors due to thermo-elastic effects. Navigation and timing information, which is part of the auxiliary data appended to the payload stream, is generated by a GPS receiver. The satellite's yaw is maintained in such way that the scanning direction is perpendicular to the momentaneous flight direction.

The AWS PFM satellite was designed to allow launch into multiple sun-synchronous orbits and no design changes are foreseen between the AWS PFM and the EPS-Sterna constellation satellites.

## 220    3.2    Launch, orbit and operation

The AWS satellite was launched Aug 16, 2024, by Space-X from the Vandenberg Space Force Base in California, USA. The Falcon-9 rocket placed AWS in a slightly elliptical orbit with a mean altitude of about 590 km. During the commissioning phase, the electrical propulsion system was used to increase the altitude and decrease ellipticity, to obtain a sun-synchronous orbit with an altitude of about 610 km. The local time of the ascending node (LTAN) is 22:38. This altitude and LTAN is

expected to be maintained as long as fuel remains for final deorbiting. The satellite complies with space debris mitigation requirements and will reenter Earth's atmosphere by natural decay within 12 years after the operations have ended. The satellite is operated from the Kongsberg Satellite Services (KSAT) operational centre in Tromsø, Norway. The NORAD ID of the satellite is 60543.

## 3.3    Downlink and dissemination of data

The satellite provides two streams of data to the ground. Global data are downlinked to KSAT's Svalbard Satellite Station (SvalSat). The selected altitude and inclination ensure that AWS is visible from the SvalSat station every revolution. The downlinked data are transmitted to Tromsø, where the raw instrument source packets for one full orbit are processed and then transmitted to EUMETSAT in Darmstadt, Germany. EUMETSAT will support the AWS mission by disseminating the full-orbit Level-1b data in a netCDF file format to end users via EUMETCast (see EUMETCast-Europe User Guide) with an anticipated

timeliness of 110 minutes. The global data reception in Svalbard, data acquisition and processing in Tromsø are part of the AWS ground-segment under a contract with ESA.

While the global data stream via EUMETCast is in most cases sufficient for data assimilation in global models, it is not adequate for nowcasting and short-range regional weather forecasting. Accordingly, AWS has also uninterrupted direct data broadcast transmission, enabling the reception of real-time data by L-band ground stations anywhere on the globe (Garcia,

2022) while the satellite is visible.

The national weather services of Finland, Norway, Denmark and Sweden, under an ESA contract and additionally supported by EUMETSAT, have set up a regional ground segment using AWS capable direct readout stations at Kangerlussuaq, Oslo





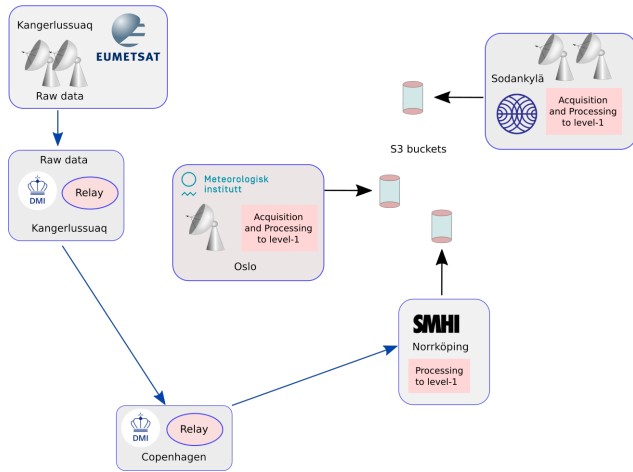

**Figure 7.** Overview of the Nordic AWS ground segment using the direct readout stations in Kangerlussuaq (Greenland), Oslo (Norway), and Sodankylä (Finland). The data received at Kangerlussuaq by EUMETSAT, will be sent to DMI and then uploaded to SMHI in Norrköping, Sweden, for final processing to level 1.

and Sodankylä (Fig. 7). This Nordic ground segment provides real-time data with a timeliness better than 15 minutes. As all three receiving stations also have other commitments, not all visible AWS passes will be received. However, pre-launch
simulations have shown that around 80% of all AWS passes will be collected. Data from all three stations will be processed to Level 1b (Sec. 3.4) and Level 1c (Sec. 3.5). File format is netCDF. Data will be available via S3 compatible object stores from Norrköping, Oslo and Sodankylä. Users will have to pull data from all three nodes in order to maximise data coverage. Contact point for getting access to these data is arctic-weather-satellite@smhi.se.

### 3.4  Level 1b

The AWS observations are provided to users primarily as Level 1b (L1b) data, i.e. calibrated and geolocated antenna temperatures. In this format, the data are reported at original boresight positions, meaning the geolocation differs between the channel groups.

The AWS radiometer is designed to facilitate the calibration process. First of all, the view against the calibration targets has been kept as clean as possible. For example, no polarizing grids or dichroic plates are used. The switching is completely
through the main rotating antenna and no other movable mirrors are involved. This inherently reduces the uncertainties in the calibration. As mentioned, AWS allows for comparably long integration times against the calibration loads, up to 37.5 and 62.5 ms for the OBCT and cold sky, respectively. In addition, these views are at hand in sampled form (at 2.5 ms resolution) making it possible to optimise the use of the calibration data.



The contribution of the far sidelobes are corrected for by the antenna pattern determination described in Sec. 4.3. A set of
pitch and roll manoeuvres were performed during the commissioning phase, for additional data on the outermost sidelobes of
the antenna system. The final L1b antenna temperatures shall be representative of an area 2.5 times the channels resolution.

For further details, see Kempe (2025). The overall responsibility of the L1b processing algorithms resides at AAC Omnisys,
while the practical implementation is performed by Elecnor Deimos, Spain. The developed processing software is applied both
by EUMESAT for the global data stream, and the Nordic AWS segment.

## 3.5 Level 1c

AWS deviates from earlier microwave cross-track scanners in not having a boresight common for all channels. As this causes
technical challenges in some assimilation systems, a processor for the remapping of data has been developed (Rydberg, 2024).
The resulting data are denoted as Level 1c (L1c). Aspects of the L1c processing are discussed below and the software itself
will be made available to end-users after testing on real data.

Ideally, the remapped representation of the data should correspond to co-located data for all channels as observed from a
common position in space. However, achieving this ideal observation is not possible. Instead, a remapping procedure can be
used to obtain common boresights and even common footprint patterns at ground level, or any other level. The remapping,
however, will have the limitation of generating data not having fully common atmospheric paths. This can be understood from
Fig. 5. There are boresight points that are close between the channel groups, but they belong to different scans. That is, even
if there is a perfect match between boresights at ground level, this position is observed from different satellite positions. This
gives a difference in the exact path through the atmosphere, as well as ground incidence angle. A second limitation is a smaller
swath width of L1c, as the start and end angles of the channel groups differ and there is a lack of overlap at the outer parts of
the swath. These features are shared with some microwave conical scanning instruments, such as GMI (Global Precipitation
Measurement Microwave Imager, Chen and Fu (2021)), MWI (MicroWave Imager) and ICI (Eriksson et al., 2020) having the
feed-horns separated in the focal plane in a similar manner as AWS. Earlier microwave cross-track radiometers have avoided
these issues by having a more complex quasi-optical system, increasing the size and cost of the instrument.

The AWS L1c processor developed (Rydberg, 2024) is based on the Backus-Gilbert (BG) footprint matching methodology
(Stogryn, 1978). In the BG methodology, a remapped value is a linearly weighted combination of footprints surrounding the
target position. The optimal weighting coefficients are obtained by minimisation of a cost function covering both the effective
noise of the remapped data and the fit to the target footprint. The derivation of the weights considers the shape of the involved
footprints, and not just boresight positions. In its general form, the BG approach allows for changing the footprint shape as
part of the remapping, but in the case of AWS it is applied to just obtain data for common boresights, at sea level with channel
group 3 defining the target positions. That is, the remapped AWS data have a geolocation following the observations of group
3, but the footprint pattern differs between the channels.

To decrease the computational burden, the L1c processor uses a set of pre-calculated channel and scan position dependent
remapping weights. Data from about 30 adjacent scans are needed for the remapping to a single scan line. The remapping
performance varies with channel and scan position, but useful remapped values are expected to be obtained for all but the last





five scan positions (of channel group 3). Systematic biases up to 1 K can not be ruled out, as the effective incidence angle differs by up to one degree between channels in the outer part of the swath. End-users of L1c data are therefore recommended to include some type of bias correction scheme if the data are used as an estimate of measurements from a single satellite location.

## 4 Pre-launch characterisation

This section summarises some of the measurements and simulations done before the launch in order to characterise the radiometer package.

### 4.1 Spectral response functions

To implement filters that exactly meet the target passband characteristics is both an expensive and time consuming effort, with low impact on the final performance (Sec. 5.3). Focus was instead given to measure the actual performance with high frequency resolution, to obtain the spectral response function (SRF) of each channel.

These measurements were performed with a continuous wave source coupled through a directional coupler, connected to the device under test and a power meter simultaneously. The set-up had a high dynamic range, was optimised to reduce standing waves and to facilitate well calibrated tests in general. The SRFs were measured for the front-ends and back-ends separately. The impact of the antenna and the feed-horns on the spectral response was not included, but should be minimal.

Obtained, end-to-end, SRFs are displayed in Fig. 8. All the responses have a relatively sharp transition from high to low response. Inside the resulting bandpass range, the responses have local maxima and minima. This variability is mainly within 3 dB. For the double sideband channels, the relative contribution of each constituent sideband must be accurately characterized. The accuracy of the obtained weights (Table 3) is not yet known and at this point the nominal values of 0.5 can be used with equal confidence.

Actual edges of the channel passbands were estimated from the measured SRFs, and were converted to the centre and bandwidths values found in Table 4 (second and third columns). To avoid ambiguity with variation inside the passbands, the bands' edges were taken as the point where the SRFs pass -6 dB. The derived band positions deviate in varying degree from their target values. For example, the measured and target width of AWS21 is 3.5 and 4.0 GHz, respectively. This deviation is not critical as the brightness temperature is close to constant over the frequency range of concern (Fig. 8). The main concern is of technical nature, a lower bandwidth makes it more challenging to meet the NEΔT target (but this is still achieved for AWS21, Sec. 6.2). The impact of deviations between target and actual band characteristics is discussed further in Sec. 5.3.

The SRFs have been incorporated in the development version of RTTOV, a widely used fast radiative transfer model (Saunders et al., 2018), and data describing the SRFs can be downloaded from https://nwp-saf.eumetsat.int/downloads/rtcoef_rttov13/mw_srf/rtcoef_aws_1_aws_srf_srf.html.



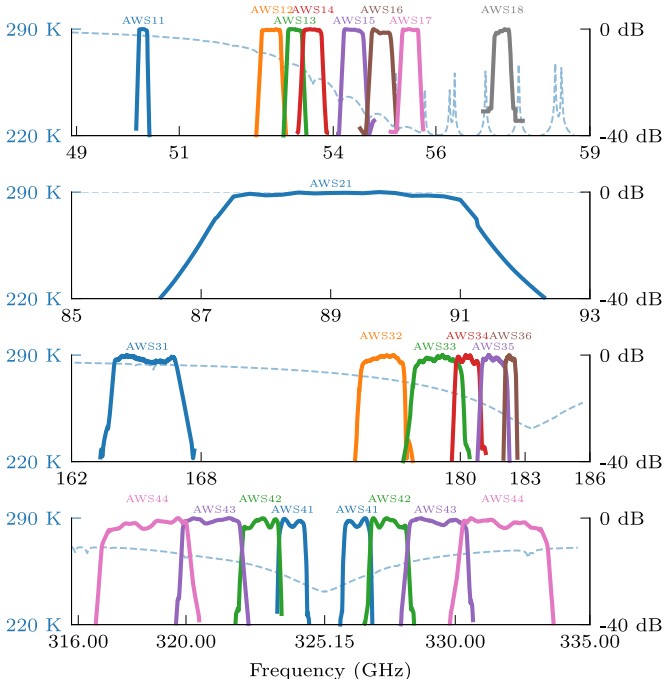

**Figure 8.** On-ground measured spectral response functions of each of the Arctic Weather Satellite's channels. The responses are normalised with respect to each peak response ($0\,\mathrm{dB}$). For context, a simulated, nadir, brightness temperature spectrum of a mid-latitude summer atmosphere is added, as blue dashed lines.

| Channel | Lower band | Upper band |
|---------|------------|------------|
| AWS41 | 0.496 | 0.504 |
| AWS42 | 0.492 | 0.508 |
| AWS43 | 0.522 | 0.478 |
| AWS44 | 0.482 | 0.517 |

**Table 3.** Relative integrated contribution of lower and upper sidebands of the 325 GHz channels.

## 4.2 OBCT

The return loss of the OBCT (Sec. 2.4) was measured using a vector network analyser. Obtained results are shown in Fig. 9.
The OBCT has an in-band return loss of $55\,\mathrm{dB}$ or better in the transverse magnetic mode, which corresponds to the orientation used for AWS. This means the electric field is perpendicular to the apex of the wedge. In the less favourable transverse electric




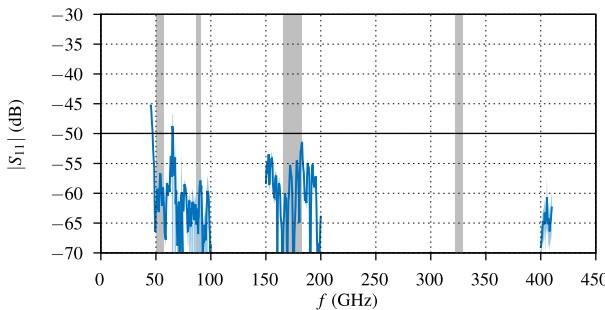

**Figure 9.** Measured return loss of AWS' OBCT for the transverse magnetic (TM) mode. Due to limitations in available test equipment, a range around 405 GHz had to be measured instead of 325 GHz.

orientation, parallel to the wedge apex, performance is only at 45 dB or better (not shown). For details on the measurement setup and methodology see Jacob et al. (2018) and Albers et al. (2024b).

### 4.3 Antenna patterns

An in-house near-field test set-up was assembled at AAC Omnisys for the characterisation of the AWS antenna pattern. The set-up consists of three main parts: a custom motorised XYZ scanner, a mechanical fixture for the instrument and an in-house designed phase/amplitude acquisition system. Synthesised frequency sources connected to custom frequency multipliers act as stimuli. The IF output from the front-end under test is down-converted and fed into the phase/amplitude acquisition system. Absorbers are used to minimise artefacts caused by multiple reflections, and the overall design aims at causing minimal flex of

sensitive cables. The set-up was found to provide excellent results in terms of both accuracy and dynamic range for the channels below 200 GHz. At 325 GHz, the desired dynamic range could not be reached due to power limitations in the transmitter, but the performance was still sufficient to validate the antenna pattern predicted by simulations.

  Post-processing of the acquired near-field data to far-field was done by IAP Bern, leveraging expertise from earlier projects. There were small misalignments of the measurement setup to the instrument, but only affecting the absolute pointing. After

manually aligning the patterns, the contour shapes could be compared (Fig. 10). The simulated and measured patterns are close to identical down to the -30 dB contours for 53 GHz. There is a good overall agreement also for the other frequencies, but deviations in details can be noted at the lower response levels. For 89 and 176 GHz the agreement is good down to -20 dB, while for 330 GHz this is valid down to -10 dB. More details can be found in Albers et al. (2024a). The minimum and maximum FWHM of the nadir response for some selected frequencies are reported in Table 4.

### 4.4 Noise

The noise performance was estimated with the radiometer package in close to final form. At the direction of "cold sky" a calibration target cooled by liquid nitrogen was used as reference. During the test, the radiometer was operated at room temperature, with nominal scan rate and integration time. The receiver noise temperatures were estimated by the variation of





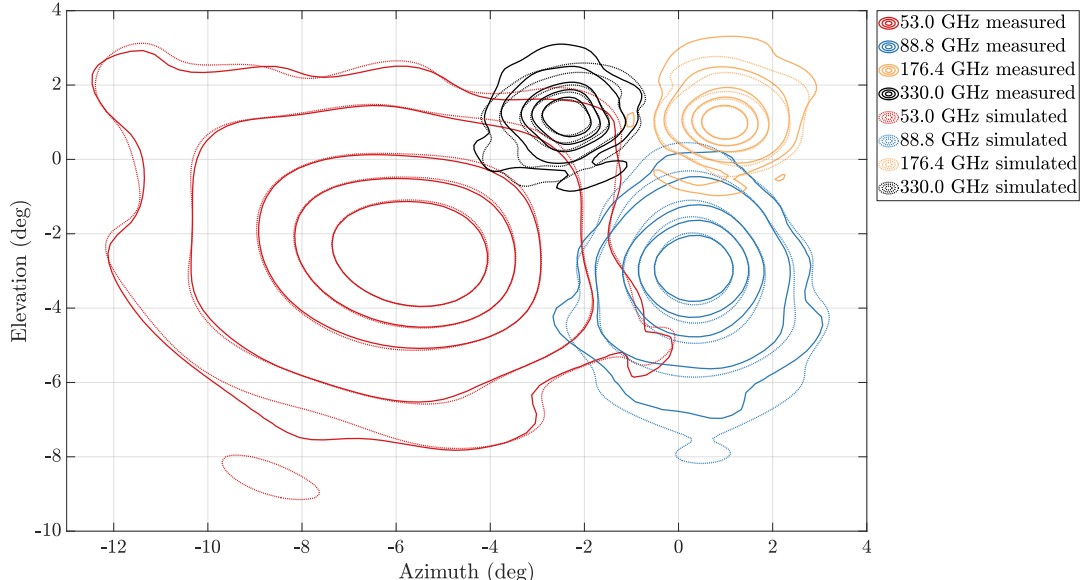

**Figure 10.** Comparison between measured (solid lines) and simulated (dotted lines) antenna responses, at 53.0 (red), 88.8 (blue), 176.4 (yellow), and 330.0 (black) GHz. The contours at -3, -6, -10, -20 and -30 dB, with respect to peak responses, are shown.

samples against the cold target. The corresponding standard deviations of noise, for a measurement against the atmosphere

(300 K assumed), are found in Table 4 (column "OG [K]"). These values can be taken as the channels' NEΔT. The NEΔT for different antenna temperatures, $T_a$ is obtained as:

$$\text{NE}\Delta\text{T}(T_a) = \frac{T_{rec} + T_a}{\sqrt{B\tau}} \tag{6}$$

where $T_{rec}$ is the receiver noise temperature of the channel, $B$ bandwidth and $\tau$ integration time.

Compared to operation in space, fewer samples towards the cold load could be obtained, resulting in more noisy calibration

data and put the values in Table 4 on the conservative side.

Most of the estimated NEΔT meet the target values in Table 1 with a margin of $> 20\%$. On the other hand, the estimated NEΔT of AWS18 is well above the target (1.0 K compared to 0.6 K). This lack of compliance was traced to a faulty component in the back-end. Since this channel is not critical for the mission (it is the one least sensitive to boundary layer weather processes, see Fig. 11) and fixing it would have delayed the launch and thus cause budget overruns, it was decided to leave

the problem unfixed. The technical problem will of course be corrected in future versions of the radiometer. There is a similar issue for AWS42, but less severe. The preliminary NEΔT of AWS42 is just above the target value (Table 1).

### 4.5 Other characterisation and space qualification

Other key radiometric performances were measured using external ambient and liquid nitrogen calibration targets, as well as a variable temperature calibration reference load (designed and manufactured by IAP). The short-term stability of the 54 and

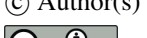



| Channel | Frequency [GHz] | Bandwidth [GHz] | Footprint FWHM max/min [km] | NEΔT OG [K] | NEΔT IO [K] | SRF Target [K] | SRF Boxcar [K] | O$_3$ [K] |
|---|---|---|---|---|---|---|---|---|
| AWS11 | 50.293 | 0.173 | 37.2/30.6 | 0.48 | 0.32 | 0.02 | 0.00 | 0.00 |
| AWS12 | 52.788 | 0.385 | | 0.33 | 0.21 | 0.19 | 0.01 | 0.00 |
| AWS13 | 53.254 | 0.294 | | 0.40 | 0.25 | 0.01 | 0.06 | 0.00 |
| AWS14 | 53.595 | 0.372 | 36.2/29.4 | 0.35 | 0.22 | 0.02 | 0.06 | 0.00 |
| AWS15 | 54.396 | 0.403 | | 0.37 | 0.22 | 0.43 | 0.34 | 0.00 |
| AWS16 | 54.933 | 0.403 | | 0.40 | 0.29 | 0.34 | 0.23 | 0.00 |
| AWS17 | 55.500 | 0.335 | | 0.50 | 0.38 | 0.01 | 0.05 | 0.00 |
| AWS18 | 57.285 | 0.316 | 35.0/28.4 | *1.18* | *0.90* | 0.10 | 0.02 | 0.00 |
| AWS21 | 89.233 | 3.748 | 20.8/17.2 | 0.21 | 0.14 | 0.01 | 0.00 | 0.00 |
| AWS31 | 165.428 | 2.979 | 11.0/9.4 | 0.36 | 0.27 | 0.11 | 0.06 | 0.34 |
| AWS32 | 176.397 | 1.985 | | 0.50 | 0.39 | 0.22 | 0.05 | 0.04 |
| AWS33 | 178.955 | 2.158 | 10.2/9.0 | 0.56 | 0.42 | 0.56 | 0.02 | 0.00 |
| AWS34 | 180.329 | 1.099 | | 0.79 | 0.58 | 0.03 | 0.14 | 0.00 |
| AWS35 | 181.543 | 1.128 | | 0.84 | 0.66 | 0.00 | 0.25 | 0.01 |
| AWS36 | 182.319 | 0.496 | 10.2/9.0 | 1.00 | 0.81 | 0.03 | 0.04 | 0.01 |
| AWS41 | 325.152±1.209 | 2× 0.941 | 11.0/9.6 | 1.60 | 1.44 | 0.04 | 0.03 | 0.30 |
| AWS42 | 325.152±2.374 | 2× 1.342 | | *1.53* | *1.53* | 0.52 | 0.24 | 0.59 |
| AWS43 | 325.150±4.200 | 2× 2.164 | 11.1/9.7 | 1.05 | 0.95 | 0.07 | 0.08 | 0.25 |
| AWS44 | 325.154±6.591 | 2× 3.021 | | 0.91 | 0.80 | 0.26 | 0.17 | 0.51 |

**Table 4.** Measured or simulated performances of the AWS-PFM channels. The values are explained and discussed as follows. Frequency and Bandwidth in Sec. 4.1, Footprint in in Sec. 4.3, NEΔT on ground (OG) in in Sec. 4.4, NEΔT in orbit (IO) and L1b in Sec. 6.2, SRF in Sec. 5.3, and O$_3$ in Sec. 5.4. The NEΔT-s of channels 18 and 42 are non-compliant, due to known reasons, and this is indicated by using italic font.

325 GHz channels was measured to reach 20 s, while for the two other two receiver chains deviations from perfect stability appeared already after 0.2 s but not in a critical manner. The dynamic range was found compliant with measuring antenna temperatures ranging from 2.7 to 300 K. The assessment of the inter-pixel error and orbital stability gave values of 0.3 and 0.2 K, respectively. The radiometric accuracy of the AWS radiometer was determined to be better than 1 K for all channels.

In line with the overall agile approach, implementation and testing applied on the development model were transferred
directly to the flight model. Changes and upgrades in the flight model were only applied in a few cases, when found necessary.



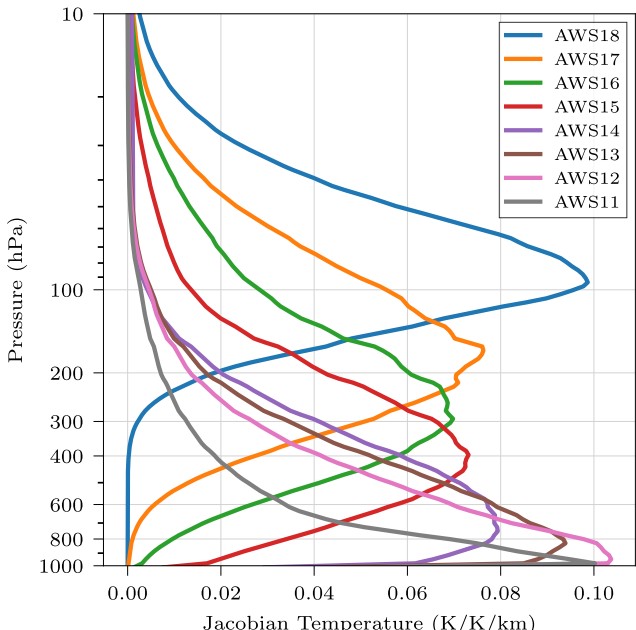

**Figure 11.** Temperature Jacobians, plotted as the change in antenna temperature for an increase in atmospheric temperature of 1 K over 1 km. A tropical scenario over a blackbody surface is assumed.

# 5 Results: Simulations

Some aspects of the measurements by AWS are exemplified by simulations. All radiative transfer simulations are done with the Atmospheric Radiative Transfer System (ARTS, Buehler et al. (2025)).

## 5.1 Clear sky

In this section it is assumed that the impact of clouds and precipitation (i.e. hydrometeors) are negligible, i.e. "clear sky" conditions. The SRFs reported in Sec. 4.1 are applied in the simulations.

The sensitivity to changes in temperature at different altitudes of the group 1 channels is illustrated in Fig. 11. These Jacobians show the derivative of the antenna temperatures to changes in atmospheric temperatures. As the centre frequency and bandwidth of the AWS temperature channels agree closely to channels on existing instruments, their Jacobians agree to 380 the ones of e.g. AMSU-A (Isoz et al., 2015).

Jacobians for channel group 3 with respect to humidity are found in Fig. 12. For the assumed conditions, the humidity Jacobians are strictly negative. That is, antenna temperatures decrease with increased atmospheric humidity. The humidity Jacobian can be positive, particularly for low peaking channels (AWS 31, 32 ...) at dry conditions above reflecting surfaces.





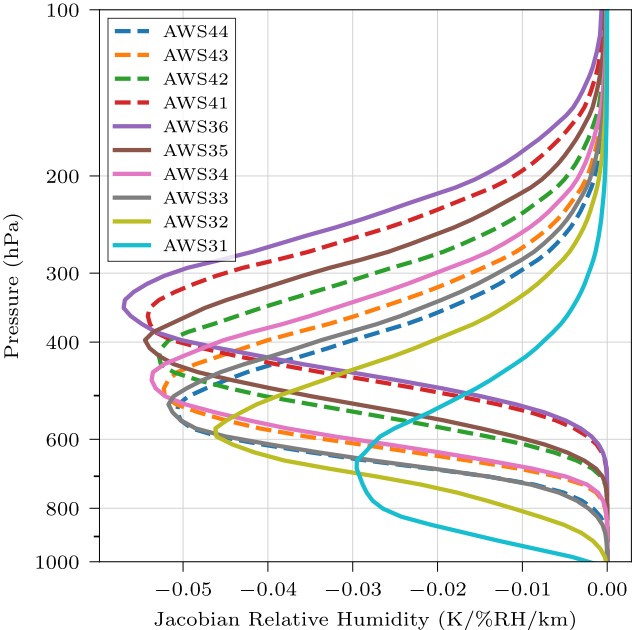

**Figure 12.** Relative humidity Jacobians, plotted as the change in antenna temperature for an increase of the relative humidity over 1 km. A tropical scenario over a blackbody surface is assumed.

The 325.15 GHz water vapour transition is of similar strength as the one at 183.31 GHz. This can be noted in Fig. 13, by

the fact that the brightness temperatures obtained around 183.31 and 325.15 GHz are similar. As a consequence, the humidity Jacobians of the AWS group 4 channels have the same magnitude as found for the four channels of group 3 closest to 183 GHz (AWS 33, 34, 35 and 36). In addition, as the channels of both groups are at similar frequency distances to their centre frequency of the transition, there is also a relatively close match in peak altitudes. In Fig. 12, the Jacobians of channels 33 and 44 are most close, but there is some variability in the matches between atmospheric scenarios. Due to the matching Jacobians, in clear

sky observations channel 41 will provide information similar as channel 36 etc. On the other hand, the impact of clouds differ clearly between the two frequency ranges as discussed in the following section.

### 5.2 All sky

The impact of atmospheric quantities in the presence of hydrometeors, i.e. "all sky", is more complex. It depends more strongly on the specifics of the atmospheric scenario at hand. Examples on all-sky Jacobians at frequencies similar to the AWS ones are

found in Birman et al. (2017) and Grützun et al. (2018).

Here, the impact of hydrometeors is instead illustrated by simulations involving three synthetic scenarios (Fig. 13). To put emphasis on the general principles, monochromatic brightness temperatures are displayed instead of channel values. The "High cirrus" case consists of an ice cloud centred at 12 km. Its vertical integral of ice content is $50 \, \text{g/m}^2$. This high cloud has





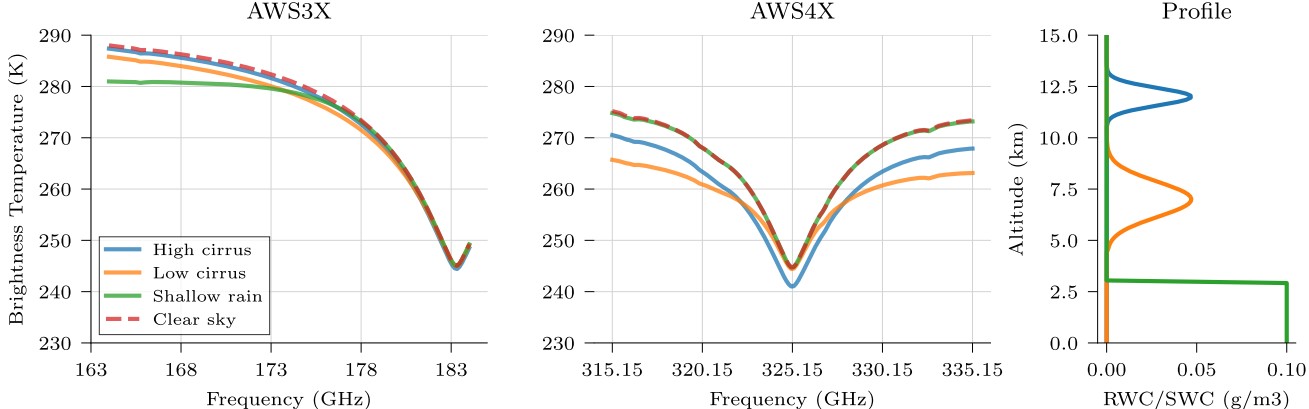

**Figure 13.** Illustration of the impact of hydrometeors inside the frequency range of AWS channels groups 3 and 4. The two left panels show top-of-the-atmosphere brightness temperatures. The right panel shows the vertical profile of rain water content (RWC) or snow water content (SWC) assumed for each simulation. Hydrometeor single scattering data are taken from Eriksson et al. (2018), with the habit "Large plate aggregate" applied for cloud ice. Particle side distributions follow Abel and Boutle (2012) and Field et al. (2007) for rain and ice, respectively.

a significant, relatively constant, impact over the full 325 GHz range. The cloud's effect around 174 GHz is a factor $\sim 6$ lower, but is also close to constant over this frequency range.

The "Low cirrus" case corresponds to an ice water content of $100 \, \text{g/m}^2$. The impact of this lower cloud varies more over both frequency ranges. This can be understood by examining the Jacobians in Fig. 12. Close to the two transition frequencies (183.31 and 325.15 GHz), the Jacobians have significant values up to about 11 km (200 hPa) showing that water vapour has high absorption up to such altitudes. As a consequence, the effect of scattering inside the cloud is more or less blocked for an observer in space. Further away from the transition frequencies, the humidity Jacobians peak lower and the gaseous absorption blocking is lower. The lower cloud gives a higher maximum impact than the high one, due to a higher ice column as well as the particle size distribution parametrisation results in larger ice particles at higher temperature.

In the "Shallow rain" case, hydrometeors exist only below about 3 km. Among the frequencies considered, a significant effect is just found in the lower end of the 174 GHz range. This is in line with the discussion around "Low cirrus" above. At a distance of 10 GHz from the transition frequencies, the impact of "Shallow rain" is larger at 173.31 compared to 315.15/335.15 GHz due to a higher low-level "continuum absorption" close to ground in the 325 GHz range.

Fig. 13 shows that the 325 GHz range adds sensitivity to the presence of ice hydrometeors. This provides information on cloud properties, similar to what is expected for ICI (May et al., 2024), but should also allow for better estimates of humidities in and below cirrus layers. The impact of such layers on observations around 183 GHz has been difficult to identify and quantify. The additional information provided by the four 325 GHz channels opens up for improved identification and correction of ice clouds (Kaur et al., 2021).





## 5.3 Impact of SRF

Simulations with different assumed SRFs were conducted to assess their impact. The values in in Table 4 (under heading "SRF"') are the maximum absolute differences between simulations across five distinct atmospheric scenarios. In both
columns, measured SRFs (Sec. 4.1) are the reference and are compared to alternative SRFs with a boxcar shape (i.e. constant within the band and zero outside). The first column ("Target") gives the difference when the boxcar SRFs are aligned with the target specifications (Table 1). These values indicate the impact of deviating from ideal responses following the target specifications, that is $< 0.6\,\mathrm{K}$.

The second column ("Boxcar"), the boxcar SRFs are placed according to the passband frequencies in the second and third
column of the same table. Assuming that the on-ground characterization constrains the positions and widths of the passbands, this serves as a conservative test of the importance of knowing the exact shape of the SRFs within the bands. The derived sensitivities are $\leq 0.3\,\mathrm{K}$.

Another view on the sensitivity to SRF is given in Fig. 14. The AWS14 Jacobian is here calculated for the measured SRF. Channel 5 of AMSU-A covers the same range, but in double sideband fashion. This results in that the oxygen transition at
53.595 GHz is inside bandwidth of the AWS channel (can be discerned in Fig. 8), but between the two sidebands of AMSU-A (see Fig. 1 in Isoz et al. (2015)). Despite this difference in SRFs, the resulting Jacobians are very similar between the instruments.

Albeit superficial, these tests show that the deviations in the passbands of AWS, both with respect to target values and other sensors, are of marginal importance. The accuracy of measured SRFs is hard to access. The results discussed above
indicate worst case impact on simulated antenna temperatures of 0.3 K, but significantly smaller for many channels. A similar conclusion, that an exact knowledge of SRFs is not critical, was reached for the ATMS sensor (Kim et al., 2014).

## 5.4 Impact of ozone

In many cases it is sufficient to only consider water vapour, oxygen and nitrogen when calculating microwave gas absorption. However, for some channels there is a significant impact of ozone (John and Buehler, 2004), and this gas is now also considered
in fast radiative transfer softwares (Turner et al., 2019). When adding channels at sub-mm wavelengths it becomes even more important to include ozone (Mattioli et al., 2019; Duncan et al., 2024a). The impact of ozone on the AWS channels is found in last column of Table 4. As was done for the SRFs sensitivities, the reported value is the maximum impact among five atmospheric scenarios. As the table indicate, there are ozone transitions inside the passbands of AWS31 and all the channels around 325 GHz. As a consequence, there will be a modelling error, albeit small, if just using a climatology mean ozone profile
in forward simulations of those channels. For more accurate results, best estimate of the local ozone profile should be applied.

## 6 Results: L1b data

A full characterisation of the L1b data quality is left for future dedicated studies, and only initial comments are provided here.




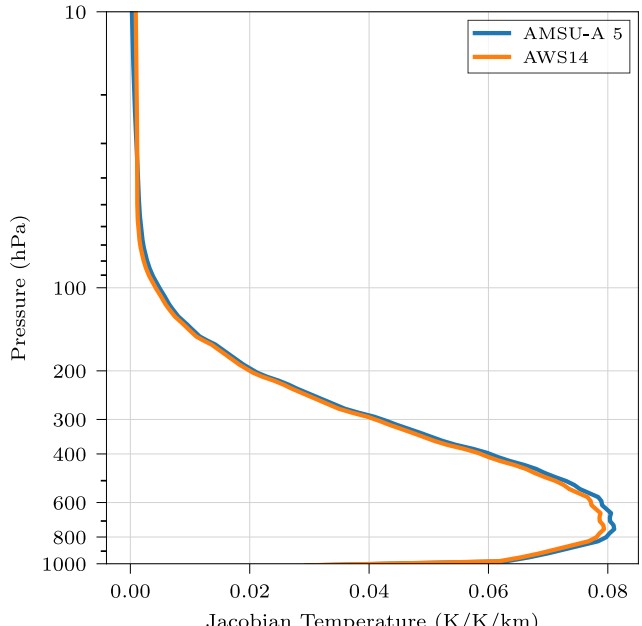

**Figure 14.** Temperature Jacobian of channels AMSU-A 5 and AWS14. Otherwise as Fig. 11

.

## 6.1 Sample L1b

A passage over the tropical cyclone Dikeledi exemplifies actual AWS measurements. This weather system formed south of
Java in the final days of 2024. Dikeledi traversed the Indian Ocean over two weeks. Fig. 15 shows satellite observations of
the storm when its centre was near the Comoros Islands, after passing northern Madagascar. As the weather system continued
over the ocean, wind speeds increased, and a day later, it made landfall as a tropical cyclone in northern Mozambique, causing
flooding, damage, and six casualties (en.wikipedia.org/wiki/Cyclone_Dikeledi). At the time of the AWS passage shown, the
storm was weaker but at a critical stage for forecasting its landfall position and strength.

As reference, the leftmost column in Fig. 15 includes observations by the Flexible Combined Imager (FCI) onboard Meteosat-
12. In the false colour representation of optical reflectivities (upper left panel) land areas come out as green or brown, while
water bodies are black. For example, lake Malawi appears as a black band stretching north-south along the 35°E meridian.
Clouds are white and light blue, and it can be seen that the storm was associated with an extensive cloud shield. However, the
cloud altitudes are not clearly depicted and optical imagery is only relevant during daytime.

The panel for Infrared (lower-left panel) and the remaining panels display measurements of thermal emission from Earth's
atmosphere or surface, presented as antenna temperatures. These radiances can be captured both day and night. For Meteosat
(lower-left panel), the window channel at 10.5 μm was selected. In cloud-free areas, radiances from this channel correspond to






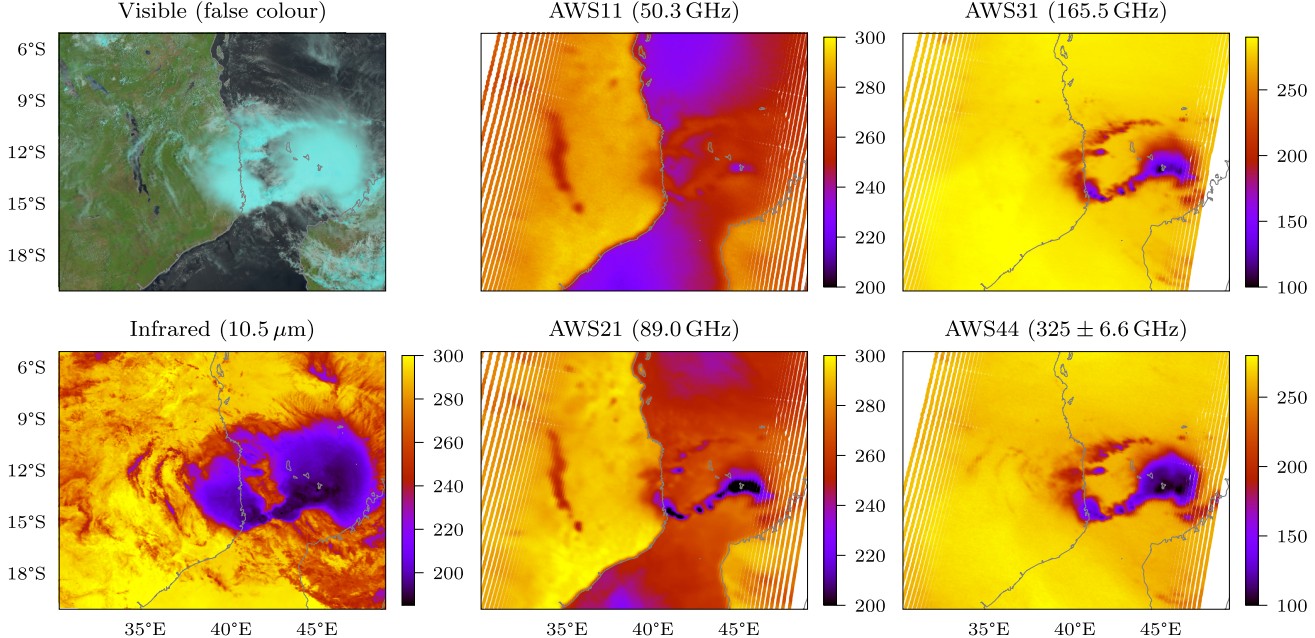

**Figure 15.** Meteosat and AWS measurements Jan 12 2025 (UTC 07:46), covering the storm Dikeledi. The coast of Mozambique is found centrally in the figure, and parts of Madagascar can be seen in the lower right corner. The unit in all colourbars is Kelvin. More details are found in the text.

.

the thermodynamic temperatures of the surface and the atmosphere just above it. In areas of optically thick clouds on the other hand the brightness temperatures of this channel correspond directly to the thermodynamic temperatures in the uppermost parts

of cloud layers. High-level clouds are most distinct, with the clouds above the storm's central part reaching altitudes around the tropopause (black areas).

    The remaining two columns of panels cover AWS, where data from one channel from each receiver chain are displayed. In all cases, the channel with the lowest (closest to the surface) sounding altitude has been selected. Starting from the right, the AWS31 and 44 channels provide information on humidity and clouds, as discussed in Secs. 5.1 and 5.2. For this warm,

humid scene there is no discernible sensitivity to the surface, not even for the AWS31 channel (Fig. 12). As plotted in Fig. 15, the most intense regions of the storm come out as black, purple and dark red. These low radiances are caused by high ice contents, generated by the intense convection found inside the storm. Compared to the infrared data, microwave radiances are more proportional to vertical integrals of ice contents than to cloud top temperatures, due to the greater penetration depth into cloud systems at longer wavelengths. In addition, while there is considerable absorption and emission by ice particles at

infrared wavelengths, the interaction with microwaves is primarily by scattering (high single scattering albedo) that decouples



the antenna temperature from the physical temperatures in the atmosphere, explaining that the value of the AWS31 and 44 data can go below 100 K.

There is an impact of the surface in the AWS11 and AWS21 data, seen as a clear contrast between radiances measured over land and water. At these frequencies, water has a considerably lower emissivity than land and therefore appears as cold. Due
to this, for AWS11 rain and liquid clouds of the storm system tend to increase the antenna temperatures over the ocean (by enhanced emission), while there is no or a decreasing effect over land (as the radiative background is warm). For AWS21, the situation is more complex. In regions with liquid clouds and moderate rain rates, AWS21 shows a higher sensitivity than AWS11, but the radiance tendencies are the same. However, in areas with strong precipitation, the antenna temperatures of AWS21 can drop below those of the ocean beneath (black areas). For this intense storm, a similar feature is observed for
AWS11 (the magenta "eye"). These low antenna temperatures indicate scattering, as seen with AWS31 and AWS44.

The Meteosat images provide a good overview thanks to their large spatial coverage, as well as revealing details of the cloud cover. However, signals with respect to precipitation are less obvious; here the situation stands out more clearly in the AWS data. For example, the black and magenta areas in AWS31 can be taken as a first-order approximation of where the most intense rain is found. Due to this relatively direct detection of precipitation, the satellite constellation of microwave radiometers constitute
the backbone of the Global Precipitation Measurement mission (GPM, Skofronick-Jackson et al. (2017)), with the GPROF system as the core tool to derive quantitative precipitation estimates from microwave antenna temperatures (Pfreundschuh et al., 2024). AWS is considered for inclusion in the extended GPM constellation, pending the release of L1b data (Rachael Kroodsma, NASA, private communication).

The response to precipitation and high ice hydrometeors masses is brought forward to most clearly differentiate the character
of microwave on one side and infrared and optical radiometric on the other side. The finer details in AWS data contain further information, primarily on, but not limited to, temperatures, humidities and liquid water contents. As such, AWS data constitute a valuable data source for numerical weather prediction. The information contained in AWS radiances is most fully exploited by assimilation systems of all-sky character (Geer et al., 2017), but as microwave observations exhibit a relatively low impact of clouds they are also of great value in assimilation restricted to clear sky conditions (Lindskog et al., 2021). In any case, AWS
data below 200 GHz are similar to existing sources and can be applied in current assimilation schemes with minor adaptations.

On the other hand, work is needed concerning the AWS 325 GHz channels. These channels have no predecessors and provide data of a new character. In existing all-sky assimilation systems, the radiative signatures of hydrometeors in 183 GHz channels mainly act to adjust the atmospheric circulation (by "4D-Var tracing", Geer et al. (2017)), while AWS' 325 GHz channels open up for also constraining ice hydrometeor amounts. This is relevant for radiative and latent heat fluxes but also allows for
better determination of the humidity in and below ice clouds. Full utilisation of this information likely requires changes of details in the assimilation systems, and for some time, the potential of the AWS 325 GHz channels could be best represented by stand-alone retrievals, in line with, e.g. Camplani et al. (2024) and May et al. (2024).



## 6.2   In-flight NEΔT

Values on the in-flight noise performance, as noise equivalent differential temperatures (NEΔTs), are found in Table 4. The
NEΔT covers contributions of thermal noise inherent in the measured signal and contributions from the instrument itself.

The in-orbit values are lower than or equal to the ones obtained on ground. Still, channels AWS18 and AWS42 exhibit
excessive noise, as discussed in Sec. 4.4. These in-orbit NEΔTs miss the requirements with about 50 and 10%, respectively.
The remaining channels meet the demands with varying margins.

## 6.3   Discussion

Short term gain fluctuations give rise to errors that exhibit correlation between measurements inside a swath. A common name
for this type of noise is "striping". Initial assessment gives that the striping in AWS data is modest. In terms of "striping ratio"
(Atkinson, 2014), it is the highest for channel groups 2 and 3 in line with the short-term stability found before launch (Sec. 4.5).
In a perspective of data assimilation, striping can still be of highest concern for the AWS temperature channels (group 1) as
"observation minus background" (O-B) is the smallest for these channels. This difference is the deviation between the actual
observation and the simulated observation based on the weather model's state, and O-B has a much higher variability for cloud
and precipitation sensitive channels making the impact of striping less apparent.

At the time of writing, final corrections for main beam antenna efficiencies, sidelobe spillover and similar features are being
determined and will be added to the operational L1b processing. Results from a preliminary setup indicate that the bias of the
AWS data is 1 K or better, using radiative transfer simulations as the reference.

The estimated performance in space has been found to in general match the on-ground characterisation. However, one
noticeable change has occurred for the 174 GHz receiver. Initial test of this receiver agreed well with the expectations, but
after an orbit correction manoeuvre performed early November (2024) it was found to behave differently. A variation of the
measured signal when sweeping the OBCT and cold sky could be noticed, a feature not previously observed. Extensive testing
has been performed and the present understanding is that a debris ended up in the feedhorn of the 174 GHz receiver during the
manoeuvre. The hypothesis is then that the debris emits radiation entering the receiver and that this unwanted emission varies
with the temperature of the debris. In its turn, the debris's temperature depends on its exposure to radiation and its thermal
inertia. Independently of the cause to the changed behaviour, its contribution has been found to be stable in time. Without
corrections, the impact on calibrated antenna temperatures, of channel group 3, manifests itself as a linear slope over the swath
of about 0.5 K.

## 7   Conclusions

The Arctic Weather Satellite (AWS) introduces a new approach inside the European Space Agency (ESA) towards designing
and developing weather forecasting-oriented missions. The overall ambition of the approach is to decrease the development
time and save costs by embracing "new space" principles. Complexity was reduced in several ways. The platform carries just





a single instrument, a cross-track microwave sounder. The number of components of the radiometer's quasi-optical system was kept as low as possible. Agile project management was applied. <The qualification philosophy focused on advancing and iterating hardware testing.

The main compromise in the payload's design is that the beams of the receiver channels are not co-aligned. This deviates from existing cross-track instruments but does not introduce any new challenge for data users as the same feature is found in conically scanning microwave radiometers. This issue is of small relevance for assimilation systems applying "super-obbing" (Duncan et al., 2024b), while others may require that a remapping of the data is performed. A software for this remapping has been developed and will be made publicly available. On the other hand, the relatively simple instrument design allowed for optimisation with respect to the calibration procedure, and the four receiver chains could be accommodated without exceeding the project's budget or deadline. In addition, one of the receivers has four channels around 325 GHz, corresponding to wavelengths around 0.9 mm, making AWS the first operational satellite using "sub-mm" wavelengths.

This article introduces the AWS satellite, focusing on the 19-channel radiometer and providing information to understand the L1b data to be disseminated. The reasoning behind the instrument's design is outlined, its main characteristics are presented, and the results from pre-launch testing are summarised. At the time of writing, the satellite is in its commissioning phase. Results obtained so far show a good consistency between on-ground and in-orbit characterisation. AWS observations are exemplified, while the detailed assessment of the in-flight performance is left for later dedicated reports. However, early testing using non-final L1b processing already indicates that AWS is providing data meeting operational standards.

Still, in a technological perspective, the AWS PFM serves mainly as a stepping stone towards the EPS-Sterna constellation. As a consequence, the satellite and the payload are undergoing unusually stringent tests during the ongoing commissioning phase to extract as many "leassons learnt" before starting the construction of the up to 18 satellites considered for the EPS-Sterna programme. EPS-Sterna, if approved by the EUMETSAT member states, will provide data with the potential to significantly improve weather forecasts (Rivoire et al., 2024; Lean et al., 2025) in an unprecedented cost-efficient manner (Varley, 2023).

*Data availability.* The options for downloading AWS data are outlined in Sec. 3.3, and links to auxiliary data resources are also found in the text.

*Author contributions.* The *primus motor* of the AWS concept is Anders Emrich. His vision was decisive for moving the AWS project through the initial critical stages before AWS became an ESA-led mission. The authors from AE to JS have worked and/or provided leadership in the construction of the AWS satellite. PE led a project inside Sweden (DICE) that can be seen as a forerunner to AWS. AT provided early support regarding weather forecasting. AC, CA and PC have led the work inside EUMETSAT regarding the dissemination of AWS L1b and preparations for EPS-Sterna. DG and VK have overseen and guided the satellite's development, and have led the commissioning phase. AWS is handled as an ESA Earth Watch mission, with VK as its project manager.



AD and VK are members of the joint ESA and EUMETSAT mission/science advisory group overlooking AWS. PE coordinated the manuscript preparation and wrote large parts of the text. AE, OA, RA, PM, BR and AD have provided direct input to the text and figures. Cited references give further insights into contributions to the development and implementation of AWS.

*Competing interests.* The authors have no competing interests to declare, besides the note that AAC Omnisys and OHB Sweden are companies with economical interest in the Sterna constellation.

*Acknowledgements.* The countries financing the AWS project are: Austria (2.4%), Canada (1.0%), Denmark (2.4%), Finland (6.0%), France (18.0%), Germany (18.0%), Norway (3.6%), Luxenbourg (2.4%), Portugal (3.6%), Spain (1.0%), Sweden (35.9%) and Switzerland (6.0%). Besides the persons listed as authors, the following employees at AAC Omnisys have contributed to the development of the AWS instrument: Stefan Andersson, Daniel Nyberg, Ulrika Krus, Mikael Krus, Ahmed Salek Brzouami, Mattias Ericson, Josefina Adebahr, Daniel Reidal, Christina Emrich, Mats Svenson, David Håkansson, Rasmus Augustsson, Steve Sahlberg, Marie Holmberg, and Mats Lindgren. Sub-
contractors to AAC Omnisys not reflected among the author list include Radiometer Physics GmbH, ACST GmbH, DA-Design Oy, Maccon GmbH, Beyond Gravity, and Space Composite Structures. The input of PE and PM to the article is supported by the Swedish National Space Agency (grant 2023-00139).





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
