# Peer review of "The Arctic Weather Satellite radiometer"

_EGUsphere, 2025_

## Author Comment (AC3)

**Response to review by anonymous referee #1**

**Opening remarks**

We warmly thank the four anonymous referees and Tim Hewison for taking the time to review our manuscript and to provide valuable feedback. As there are commonalities between several of the reviews, we start with some general remarks. To begin, we emphasise that our goal is not to encompass the entire AWS mission. First of all, this would be very challenging to cover within a standard manuscript length and would approximately double the number of co-authors. For example, the primary objective of AWS is numerical weather prediction (NWP), and addressing the aspects and applications of AWS within this area could be a manuscript in itself. The manuscript's aim is instead to provide the necessary information to understand the design of the AWS radiometer and to utilise the L1b data from this instrument. In the revision, we focus on improving the text around these aspects based on the provided feedback, as well as adding some new information.

A related question is how much in-orbit characterisation to include. Here, we hope to have an understanding of the difficulty of compiling the manuscript at the same time as the team is preoccupied with the satellite's commissioning phase. The initial aim was to submit the manuscript in 2024. In particular, the sudden deviating behaviour of the 174 GHz receiver (Sec. 6.3) caused significant concern and resulted in a substantial delay in the manuscript. Nevertheless, our approach is to include some initial basic results, primarily to indicate that the findings from the on-ground tests appear to be valid. We have added a sentence to exemplify this further and on the same time indicate the range of aspects that has to be considered. We avoid going further to leave room for one or several upcoming articles that are entirely focused on in-orbit testing. In addition, to fully cover the in-orbit testing would again require a considerable extension of the list of authors. This work is ongoing and far from complete. At least one update of the L1b processing algorithm is foreseen.

In summary, we find it reasonable to focus on the development of the instrument up to the launch. On this side, we think the manuscript is already more information-rich than usual. This brings us to an unstated objective. It is already difficult to find in the open literature the relevant background information about the satellite instruments we use for research. The trend towards new space and more substantial commercialisation risks making the situation worse; with this manuscript, we aim to demonstrate that this need not be the case.

The replies below refer to the revised version of the manuscript we have prepared.

**Replies on referee's comments**

- "The hyperlink/URL does not appear to be functional."
  - URL for new location of AWS SRFs is updated.
- "Figures 8, 12, 13, 14, 15, and 16: Axis values are repeated and not easily interpretable."
  - As mentioned in author comment 1 (https://doi.org/10.5194/egusphere-2025-1769-AC1), the issues with the figures were due to a conversion problem and should now be resolved.
- "The text refers to the "new space" philosophy/principles. It would be helpful to include a reference or add one or two sentences briefly explaining this".
  - A clarifying sentence on new-space has been added to the Introduction. It connects with following sentences on how the approach was applied by ESA.
- "Minor textual corrections"
  - Textual corrections are applied as suggested.

---

## Author Comment (AC4)

**Response to review by anonymous referee #2**

**Opening remarks**

We warmly thank the four anonymous referees and Tim Hewison for taking the time to review our manuscript and to provide valuable feedback. As there are commonalities between several of the reviews, we start with some general remarks. To begin, we emphasise that our goal is not to encompass the entire AWS mission. First of all, this would be very challenging to cover within a standard manuscript length and would approximately double the number of co-authors. For example, the primary objective of AWS is numerical weather prediction (NWP), and addressing the aspects and applications of AWS within this area could be a manuscript in itself. The manuscript's aim is instead to provide the necessary information to understand the design of the AWS radiometer and to utilise the L1b data from this instrument. In the revision, we focus on improving the text around these aspects based on the provided feedback, as well as adding some new information.

A related question is how much in-orbit characterisation to include. Here, we hope to have an understanding of the difficulty of compiling the manuscript at the same time as the team is preoccupied with the satellite's commissioning phase. The initial aim was to submit the manuscript in 2024. In particular, the sudden deviating behaviour of the 174 GHz receiver (Sec. 6.3) caused significant concern and resulted in a substantial delay in the manuscript. Nevertheless, our approach is to include some initial basic results, primarily to indicate that the findings from the on-ground tests appear to be valid. We have added a sentence to exemplify this further and on the same time indicate the range of aspects that has to be considered. We avoid going further to leave room for one or several upcoming articles that are entirely focused on in-orbit testing. In addition, to fully cover the in-orbit testing would again require a considerable extension of the list of authors. This work is ongoing and far from complete. At least one update of the L1b processing algorithm is foreseen.

In summary, we find it reasonable to focus on the development of the instrument up to the launch. On this side, we think the manuscript is already more information-rich than usual. This brings us to an unstated objective. It is already difficult to find in the open literature the relevant background information about the satellite instruments we use for research. The trend towards new space and more substantial commercialisation risks making the situation worse; with this manuscript, we aim to demonstrate that this need not be the case.

The replies below refer to the revised version of the manuscript we have prepared.

**Replies on referee's comments**

General comments:

- "I found the order of the sections to be awkward, and suggest following the order closer to the ESA SMOS paper ..."
  - We have decided to not follow this suggestion. The proposed rearrangement builds on, at least partly, that the full mission is treated, which is not the case as described above. Section 6.1 is included only to exemplify the AWS data, in particular to give a first glimpse from the novel 325 GHz channels. We also refer to that this subject has not been brought up by the other four reviewers.

Specific comments:

- "The paper's title should include the word Mission to go with Radiometer"
  - See Opening remarks above.
- "Merge radiometer background (Sect. 2.1) with intro section (Sect. 1.0). It wasn't until 2.1 that I learned why Artic is in the name."
  - Thanks for this suggestion, we agree that having the background and "history" behind AWS in the Introduction is better. We have moved Section 2.1 to the introduction with minor adaptations to make it flow with the rest of the introduction, at the same time as text has been incorporated in response to a comment from another referee.
- "I see a statement on Line 368 of radiometric accuracy, but no target in Table 1."
  - Table 1 is not meant to give a complete coverage of the long list of all requirements.
- "Per the ATBD, L1b is brightness temp., but the paper seems to consistently call it antenna temperature (traditionally ant. temp. is L1a, but AWS doesn't seem to offer it)."
  - An observant remark! We avoid going into a discussion of the correct term for calibrated data, and approach the question from a pragmatic standpoint. Since the submission we have noticed that this question has generated confusion even inside the broader AWS team (i.e. including data end users), and we have adopted the nomenclature to the one of ATBD and L1b data. That is, antenna temperature is now replaced with brightness temperature.
- "Sect. 4.0 Pre-launch Characterization: I didn't see any discussion on non-linearity. The only thing I could find in the ATBD was a detector nonlinearity correction without mention on how it was derived. Also, it doesn't look like AWS radiometer went through TVac calibration where an instrument-level non-linearity correct is derived? While I'm an advocate of the New Space approach, the non-linearity is very hard to derive on-orbit ..."

- Comments on these points are now found in Secs. 2.2 and 4.5.
- "Sect. 4.5: Can you include references or more details on this section? There seems to be all results with no information on how it was calculated."
  - Sec 4.5 has been rewritten and hopefully better clarifies the approach and ambition of the tests.
- "NEDT consistency (Sect. 2.3, 4.4, & 6.2): It doesn't seem that the comparison of the various stated NEDTs are consistent. Just looking for more details and not re-analysis. 6 is the intrinsic NEDT equation, but there are other sources of NEDT contribution: ..."
  - A rigorous assessment of NEDT is surprisingly difficult. The contribution of gain variations (striping) is especially challenging. The AWS team is fully aware of these issues, and it is acknowledged that the NEDT values presented in this manuscript are approximative. However, there seems also be some misunderstanding and we have done some changes around Eq 6 for better clarity.
  - On request from several of the referees, the way NEDTs have been derived is now described (in Sec 4.4). The same calculation approach was used for on-ground and in-orbit data, clarified in Sec 6.2.
- "Sect. 2.6 Scan Sequence: The ATBD mentioned two potential cold sky sectors (ATBD Sect. 3.6.1) and that "the cold sky measurement depends on the orbit and occur before or after the earth scene." What was the final result?"
  - The two different cold sky views are mainly of interest for Sterna, with satellites in different orbits. For AWS only a single view can be used. As this is taken from ATBD, no change in the manuscript.
- Sect. 3.1: "What, if any, thermal control of the radiometer is there?"
  - Sec 2.1 has been extended with a sentence addressing the question.
- Sect. 3.1: "Can you add something on geolocation target accuracy and/or point to your sensitivity study in Table 2 of AWS-OMN-RP-0002 Issue C? This report seems to have more info. than the referenced AWS-SMHI-RP-0002 Issue A?"
  - In lack of values on the possible errors of the angles of concern, including the table seems not motivated at this point. In addition, the geolocation does not only depend on those angles, also e.g. timing issues are of concern. See further the next answer.
- Sect. 3.1: "What on-orbit verification (e.g., Coastline Inflection Point technique) will be used to tune the geolocation parameters?"
  - In line with the Opening remarks, we don't go into details of the geolocation accuracy. There will be a dedicated journal article on the subject.

- Sect. 3.1: "What pre-launch measurements were made to confirm pointing knowledge? The antenna pattern is a start, but doesn't include the alignment/transform between the instrument and the spacecraft LVLH control. Is it just the close placement of the star trackers to the payload and use a post-launch empirical correction based on CIP?"
  - The efforts made are now outlined in Sec. 4.5.
- Sect. 3.4: "Consider adding a less detailed version of the ATBD Fig. 6 "Overview of the AWS instr. signal proc. chain" be included that allows the activities in the Pre-launch section be tied to the calibration algorithm?"
  - We understand the interest in the actual calibration algorithm, but still think simply referring to Kempe (2025) is the best solution for the manuscript. Including a simplified figure can cause confusion of the actual algorithm, and would still require a significant extension of the manuscript to explain the figure.
- Sect. 3.4: "Regarding Line 260 starting with "The final L1b…": Is this 2.5 times the channel's FWHM projection on the surface?"
  - Yes. Text changed.
- Sect 4.1: "line 307: "should be minimal" Is there any reference for this statement? …"
  - Yes, an unclear statement. The text has been changed to be more informative then just saying minimal. New information has also been added to clarify that front- and back-end combinations were also measured, but over a lower dynamic range.
- Sect. 5.3: "Please add more info. on the "five distinct atmospheres" or a reference."
  - We now explicitly mentioned the five scenarios and a reference to the data at the start of section 5.3.
- Sect. 5.3: "Regarding AWS14 passband crossing over the absorption line, what spectral sampling did you use?"
  - Thanks for this keen observation. The frequency sampling in our simulations was not dense enough to properly resolve the O2 transition covered by AWS14. Thanks to this comment, the sampling is substantially increased to properly resolve this line, including a sample point right on the transition. Figure 14 (Temperature Jacobian of AMSU-A and AWS14) and Table 4 (simulated channel performances) have been updated accordingly. We see no discernible difference in Figure 14, and very small differences for AWS14 in Table 4:
    - Max difference measured boxcar is changed from 0.06K to 0.05K

- - After addressing comments by CC1 on how bandwidths are measured, the final value in Table 4 is 0.03K
    These changes do not change any conclusions in the text.
- Sect. 5.3: "How do RTTOV-folks handle it?"
  - How AWS is handled inside RTTOV is not inside the manuscript's scope.
- Sect. 5.3: "I'm not clear on the take away or point of the last paragraph. [...] I think you should emphasize that you're saying that any residual SRF uncertainties are marginal. Using the ATMS has an example isn't the same in my opinion because they had the strict requirements in place, so the SRF was fairly close to the boxcar."
  - As the reviewer helpfully points out, the ATMS comparison in the section's concluding paragraph adds confusion to the message and is therefore removed. We focus on saying that 1) deviations from target specifications are small and 2) that measured SRF gives even smaller differences over the adjusted boxcar SRF.
- Sect. 5.4: "Is the impact in Table 4 the difference between zeroing out the O3 or using the climatology mean O3 profile? That is, are the numbers in Table 4 the residual error of using the mean O3 profile?"
  - Table 4 reports the difference having O3 and zeroing it out, across the same five atmospheres as used for the SRF analysis. We have clarified this in Section 5.4.

---

## Author Comment (AC5)

**Response to review by anonymous referee #3**

**Opening remarks**

We warmly thank the four anonymous referees and Tim Hewison for taking the time to review our manuscript and to provide valuable feedback. As there are commonalities between several of the reviews, we start with some general remarks. To begin, we emphasise that our goal is not to encompass the entire AWS mission. First of all, this would be very challenging to cover within a standard manuscript length and would approximately double the number of co-authors. For example, the primary objective of AWS is numerical weather prediction (NWP), and addressing the aspects and applications of AWS within this area could be a manuscript in itself. The manuscript's aim is instead to provide the necessary information to understand the design of the AWS radiometer and to utilise the L1b data from this instrument. In the revision, we focus on improving the text around these aspects based on the provided feedback, as well as adding some new information.

A related question is how much in-orbit characterisation to include. Here, we hope to have an understanding of the difficulty of compiling the manuscript at the same time as the team is preoccupied with the satellite's commissioning phase. The initial aim was to submit the manuscript in 2024. In particular, the sudden deviating behaviour of the 174 GHz receiver (Sec. 6.3) caused significant concern and resulted in a substantial delay in the manuscript. Nevertheless, our approach is to include some initial basic results, primarily to indicate that the findings from the on-ground tests appear to be valid. We have added a sentence to exemplify this further and on the same time indicate the range of aspects that has to be considered. We avoid going further to leave room for one or several upcoming articles that are entirely focused on in-orbit testing. In addition, to fully cover the in-orbit testing would again require a considerable extension of the list of authors. This work is ongoing and far from complete. At least one update of the L1b processing algorithm is foreseen.

In summary, we find it reasonable to focus on the development of the instrument up to the launch. On this side, we think the manuscript is already more information-rich than usual. This brings us to an unstated objective. It is already difficult to find in the open literature the relevant background information about the satellite instruments we use for research. The trend towards new space and more substantial commercialisation risks making the situation worse; with this manuscript, we aim to demonstrate that this need not be the case.

The replies below refer to the revised version of the manuscript we have prepared.

**Replies on referee's comments**

- "There has been considerable recent interest and activity in the area of earth observing small satellites that typically require sacrifices in performance, capability, and/or reliability in order to reduce costs and facilitate their implementation in an effective and expeditious manner. This paper does not adequately mention or reference prior art or previous work in this area ..."

  - These are highly important considerations that have been discussed in the work leading up to AWS. For example a review of cubesat mission was made. However, it is difficult to discuss the matters in a rigorous manner as the details and the final performance of those small missions are not clearly described in the open literature. However, there is a fundamental difference between AWS and cubesat missions (as far as we understand). While sacrifices in performance seem to be accepted for those very small missions, AWS aims to be at least similar to the state-of-the-art missions in terms of core performance.

  - We tried to avoid this complicated question, but the referee is correct, the relationship to prior work shall be clear. The impressive technical feat behind cubesat missions has acted as inspiration for AWS, but the actual roots of the mission is rather find in some national (Swedish led) satellite mission. There is a new paragraph in the Introduction, indicating these links.

- "Regarding the technical content of the paper, many details are given in the paper, but I think there are a few basic pieces of information that are missing that would be of keen interest to the readership. For example, what is the mass, volume, power consumption, and data rate of the instrument (and even better, simple comparisons to what is flying now or is planned to fly)? There is some of this kind of information presented for the satellite bus, but instrument parameters would be more meaningful."

  - Thanks for pointing out this neglection. Numbers on the mass, power consumption and data rate have been added (end of Sec. 2.1).

- "The instrument does not include "traditional" channels near 24 and 31 GHz due to the size of the reflector that would be needed. This is a reasonable design trade, but those are very important channels for the retrieval of total precipitable water – it would be useful to simply discuss the impact of this omission and how it might be mitigated."

  - There is no mitigation on the AWS platform for the lack of channels at 24 and 31 GHz. However, a discussion of this omission is far from trivial, and we select to not include such a one. In short, it is clear that AWS would have provided more information on total precipitable water if also having those channels, given the budget and time needed to implement the channels without any negative effect on the other channels. If the issue

instead is looked upon with actual budget and time as constraints, AWS would have become a poorer instrument by including 24 and 31 GHz as it would have demanded significant compromises for the other channels, including a full removal of the 325 GHz receiver chain and possible also some other. Or more likely, if suggested with 21 and 31 GHz the budget and the technical risks would likely been seen as too large and AWS would not had been realized at all.

- "Another design choice appears to be to fly the satellite at a lower altitude than current operational sensors (600 km versus 817 km). This of course yields better spatial resolution for a given reflector size, but at the cost of substantial footprint broadening at larger instrument scan angles. The spatial resolution at the scan edges (55 degrees) could be much too coarse for effective operational use. Some discussion of this point would be helpful – what are the AWS scan-edge resolutions and how do these compare with present systems, for example? And how does this impact the planned approach of spatially combining/aligning the footprints for the various bands?"

    - This is a fair comment, but we again select to not extend the discussion due to the complexity of the question and limit the information to: An altitude of 600 km has been targeted as a good compromise between brightness size requirement, ground coverage, launch cost and end-of-life deorbiting considerations.
    As this comment indicate, there are multiple considerations when selecting the orbit altitude, including ones around launch and deorbiting that largely excluded orbits around 800 km. To this can be added that the many NWP centers, when assimilating data from microwave cross-track scanners, anyhow exclude data from the outer parts of the swath (as far as we understand).

- "Another compromise is the choice (reasonably) of a constant scan velocity versus and non-constant scan velocity (whereby the scan is slower over the earth and faster away from the earth, so that the integration time for earth-viewing footprints is longer, thus noise is lower). The penalty paid for this is approximately sqrt(2) in noise amplification (assuming scan accelerations consistent with current operational sounders). Again, some discussion of the regret of this would be useful, especially in light of the profound impact of the radiance noise in numerical weather prediction applications."

    - A short discussion has been added at the beginning of Sec 2.5.

- "The terms "inter-pixel error" and "orbital stability" are used without definition – what are these and how were they quantitatively assessed?"

    - The choice of these words likely indicated a standard assessment of these aspects, that is not correct. Sec 4.5 has been rewritten and hopefully better clarifies the approach and ambition of the tests.

- "I believe the L-band satellite communications transmitter frequency (1.7 GHz) falls within the IF bands of the high-frequency receivers - was any prelaunch testing done to ensure electromagnetic compatibility of the spacecraft hardware and the radiometer? Does the radiometer noise measured on-orbit increase when the communications transmitter is on?"
    - This possible interference has been considered and tested, both on ground and in orbit. Comments about this have been added to Secs. 2.2 and 6.3.

---

## Author Comment (AC6)

**Response to review by Tim Hewison**

**Opening remarks**

We warmly thank the four anonymous referees and Tim Hewison for taking the time to review our manuscript and to provide valuable feedback. As there are commonalities between several of the reviews, we start with some general remarks. To begin, we emphasise that our goal is not to encompass the entire AWS mission. First of all, this would be very challenging to cover within a standard manuscript length and would approximately double the number of co-authors. For example, the primary objective of AWS is numerical weather prediction (NWP), and addressing the aspects and applications of AWS within this area could be a manuscript in itself. The manuscript's aim is instead to provide the necessary information to understand the design of the AWS radiometer and to utilise the L1b data from this instrument. In the revision, we focus on improving the text around these aspects based on the provided feedback, as well as adding some new information.

A related question is how much in-orbit characterisation to include. Here, we hope to have an understanding of the difficulty of compiling the manuscript at the same time as the team is preoccupied with the satellite's commissioning phase. The initial aim was to submit the manuscript in 2024. In particular, the sudden deviating behaviour of the 174 GHz receiver (Sec. 6.3) caused significant concern and resulted in a substantial delay in the manuscript. Nevertheless, our approach is to include some initial basic results, primarily to indicate that the findings from the on-ground tests appear to be valid. We have added a sentence to exemplify this further and on the same time indicate the range of aspects that has to be considered. We avoid going further to leave room for one or several upcoming articles that are entirely focused on in-orbit testing. In addition, to fully cover the in-orbit testing would again require a considerable extension of the list of authors. This work is ongoing and far from complete. At least one update of the L1b processing algorithm is foreseen.

In summary, we find it reasonable to focus on the development of the instrument up to the launch. On this side, we think the manuscript is already more information-rich than usual. This brings us to an unstated objective. It is already difficult to find in the open literature the relevant background information about the satellite instruments we use for research. The trend towards new space and more substantial commercialisation risks making the situation worse; with this manuscript, we aim to demonstrate that this need not be the case.

The replies below refer to the revised version of the manuscript we have prepared.

**Replies on referee's comments**

- "L175: The scan rate is constant, giving an along-track distance between footprints of about 9.0 km"
    - We just wanted to give a rough number and it was a mistake to write 9.0 (instead of just 9). And we probably missed to include the scaling down to the ground altitude. Thanks for the correction, we have changed to your value.
- "I suggest to make it clear than the operational altitude of AWS has been a fairly constant 599km since 2024-12-01."
    - Changed from "altitude of about 610 km" to "a semi-major axis altitude of 599 km".
- "L315: Please highlight and justify the departure from the usual convention is to report bandwidths between 3dB points (not 6dB)."
    - As there is considerable variation inside the passbands, we selected -6dB (with respect to peak response) to be conservative. However, we agree that it is strange to not use the standard -3dB. Therefore, we have changed our method and motivation in the text. We now normalize each SRF with respect to the average response values between the peak-normalised -3dB points. Using this band average as reference, we take the usual -3dB bandwidth. This is described in Sec 4.1. The measured "Boxcar" difference values in Table 4 are updated.
- "L344: The minimum and maximum FWHM of the nadir response for some selected frequencies are reported in Table 4. – Add mean"
    - A good suggestion, adopted.
- "L349: Over what period were the standard deviations evaluated?"
- "L361 + L509: How was the In-Orbit NEDT evaluated? Is this based on Deep Space on OBCT views?"
    - On request from several of the referees, the way NEDTs have been derived is now described (in Sec 4.4). The same calculation approach was used for on-ground and in-orbit data, clarified in Sec 6.2.
- "L364: How is the short-term stability defined?"
- "L367: How were the inter-pixel error and orbital stability quantified?"
    - The choice of these words likely indicated a standard assessment of these aspects, that is not correct. Sec 4.5 has been rewritten and hopefully better clarifies the approach and ambition of the tests.
- "L419: What are the 5 atmospheric scenarios used to define the SRF sensitivity?"

- We agree with this comment and a similar one by RC2 that this should be mentioned. We now explicitly mention the five scenarios and add a reference to the data source at the start of section 5.3.

- "L420: How is the SRF sensitivity actually quantified in Table 4? Is this the mean difference between the brightness temperatures simulated by ARTS with the actual SRF and with the boxcar approximation? The difference will be scene dependent – can you quantify its variance?"

  - Thanks for this comment. In our view, these questions are addressed in the manuscript. How SRF sensitivity in Table 4 is quantified is introduced at the beginning of Section 5.3, and O3 sensitivity in Section 5.4. Here, it is also described that the values are the maximum absolute difference between measured SRF and "Target" or "Boxcar" over five atmospheric scenarios (now also specified to FASCOD according to the previous comment).

  - The difference is indeed scene-dependent. However, only the atmospheric profile is changed according to the five Fascod atmospheres. These atmosphere profiles are intended to reflect a common state for tropical, midlatitude, or sub-arctic regions during summer or winter. In the interest of not cluttering Table 4 too much, and since there are no distinctive scenarios included, we think that a worst-case value across these simulations serves the analysis of SRF performance best.

  - However, to still address the question, we include the individual differences across cases for AWS15 and AWS42 (the two channels with the largest difference in the Boxcar column) in this reply:

|  |  |  | Ta abs(MEASURED-BOXCAR) | Ta abs(MEASURED-BOXCAR_FROM_MEASURED) |
| --- | --- | --- | --- | --- |
| channel | AbsSpeciesCase | FascodAtm |  |  |
| AWS15 | RTTOV_v13x | midlatitude-summer | 0.36 | 0.12 |
|  |  | midlatitude-winter | 0.26 | 0.09 |
|  |  | subarctic-summer | 0.27 | 0.10 |
|  |  | subarctic-winter | 0.21 | 0.07 |
|  |  | tropical | 0.43 | 0.14 |
| AWS42 | RTTOV_v13x | midlatitude-summer | 0.50 | 0.22 |
|  |  | midlatitude-winter | 0.45 | 0.17 |
|  |  | subarctic-summer | 0.38 | 0.18 |
|  |  | subarctic-winter | 0.39 | 0.15 |
|  |  | tropical | 0.52 | 0.22 |

---

## Author Comment (AC7)

**Response to review by anonymous referee #4**

**Opening remarks**

We warmly thank the four anonymous referees and Tim Hewison for taking the time to review our manuscript and to provide valuable feedback. As there are commonalities between several of the reviews, we start with some general remarks. To begin, we emphasise that our goal is not to encompass the entire AWS mission. First of all, this would be very challenging to cover within a standard manuscript length and would approximately double the number of co-authors. For example, the primary objective of AWS is numerical weather prediction (NWP), and addressing the aspects and applications of AWS within this area could be a manuscript in itself. The manuscript's aim is instead to provide the necessary information to understand the design of the AWS radiometer and to utilise the L1b data from this instrument. In the revision, we focus on improving the text around these aspects based on the provided feedback, as well as adding some new information.

A related question is how much in-orbit characterisation to include. Here, we hope to have an understanding of the difficulty of compiling the manuscript at the same time as the team is preoccupied with the satellite's commissioning phase. The initial aim was to submit the manuscript in 2024. In particular, the sudden deviating behaviour of the 174 GHz receiver (Sec. 6.3) caused significant concern and resulted in a substantial delay in the manuscript. Nevertheless, our approach is to include some initial basic results, primarily to indicate that the findings from the on-ground tests appear to be valid. We have added a sentence to exemplify this further and on the same time indicate the range of aspects that has to be considered. We avoid going further to leave room for one or several upcoming articles that are entirely focused on in-orbit testing. In addition, to fully cover the in-orbit testing would again require a considerable extension of the list of authors. This work is ongoing and far from complete. At least one update of the L1b processing algorithm is foreseen.

In summary, we find it reasonable to focus on the development of the instrument up to the launch. On this side, we think the manuscript is already more information-rich than usual. This brings us to an unstated objective. It is already difficult to find in the open literature the relevant background information about the satellite instruments we use for research. The trend towards new space and more substantial commercialisation risks making the situation worse; with this manuscript, we aim to demonstrate that this need not be the case.

The replies below refer to the revised version of the manuscript we have prepared.

**Replies on referee's comments**

- "Considering that the instrument has been on orbit for several months at the time this manuscript was submitted, I would expect more comprehensive on-orbit performance information than just the individual channel noise estimates. It would be useful to see these, along with comparisons to heritage instruments such as ATMS and MHS, to demonstrate the viability of the "new space", small satellite approach. Otherwise, this is an excellent and informative manuscript for users of AWS data."

    - We understand and appreciate the wish to learn more about AWS, but we argue against this extension as explained in the opening remarks.

- "Figure 13: It would be helpful to illustrate the locations of the AWS 3x and 4x channel bands in these plots."

    - We agree and have updated Figure 13 accordingly.

- "Line 215: It is mentioned that the instrument is designed to minimize geolocation error here, but no geolocation accuracy statistics are presented. Especially considering the novel feedhorn arrangement, this would be particularly useful to assess in this manuscript."

    - We don't claim that AWS is designed to minimize geolocation at an overall level, just that the placement of the star trackers is selected considering the thermo-elastic effects. In line with the Opening remarks, we don't go into details of the geolocation accuracy. There will be a dedicated journal article on the subject.

- "Lines 365-368: A lot of information is given here about parameters that are important for real-world radiometric accuracy and at the end of the paragraph, it is stated that the accuracy is better than 1 K for all channels. However, how exactly was this determined (e.g., how was non-linearity assessed, was the on-orbit thermal cycle modeled during the calibration testing, what on-orbit maneuvers were used to calculate spillover)?"

    - Sec 4.5 has been rewritten and hopefully better clarifies the approach and ambition of the tests.

- "Line 434: I believe this should be The impact of measured SRFs is hard to *assess*."

    - Yes, that was a mistake. However, the paragraph is now rewritten, in response to a comment by another referee.

- "Table 4: Are the on-orbit NEDT values also scaled to a 300 K scene? I would assume so but it is not explicitly stated. Also, are they derived from the variance of counts in the warm calibration sector, cold calibration sector, or both (and what receiver temperature was assumed)? Also, since it is mentioned in the Discussion section, it would be nice to have the striping index added to this

table, since that is a standard performance parameter for  microwave radiometers."

- On request from several of the referees, the way NEDTs have been derived is now described (in Sec 4.4). The same calculation approach was used for on-ground and in-orbit data, clarified in Sec 6.2. The scaling to 300K was mentioned, but is now made clearer in Table 4. An assessment of striping we leave for future publications, as mentioned in the Opening remarks.

---

## Author Response (AR2)

The technical correction has been implemented.